

# Dynamics of salt intrusion in the Mekong Delta; results of field observations and integrated coastal-inland modelling

Sepehr Eslami[1,2], Piet Hoekstra[1], Herman W.J. Kernkamp[3], Nam Nguyen Trung[4], Dung Do Duc[4], Hung
Nguyen Nghia[5], Tho Tran Quang[4], Arthur van Dam[3], Stephen E. Darby[6], Daniel R. Parsons[7], Grigorios
Vasilopoulos[7], Lisanne Braat[8], Maarten van der Vegt[1]

[1]Department Physical Geography, Faculty of Geoscience, Utrecht University, Utrecht, 3584 CB, the Netherlands
[2]Marine and Coastal Systems Unit, Deltares, Delft, 2629 HV, the Netherlands
[3]Software Department, Deltares, Delft, 2629 HV, the Netherlands
[4]Southern Institute for Water Resources Planning (SIWRP), Ho Chi Minh City, Ward 3 72710, Vietnam
[5]Southern Institute of Water Resources Research (SIWRR), Ho Chi Minh City, 656 Võ Văn Kiệt, District 5, Vietnam
[6]Geography and Environment, University of Southampton, Southampton SO17 1BJ, UK
[7]Department of Geography, Environment and Earth Sciences, University of Hull, Hull HU6 7RX, UK
[8]Department of Geology and Planetary Sciences, California Institute of Technology, CA 91125, USA

*Correspondence to*: Sepehr Eslami (sepehr.eslamiarab@deltares.nl)

**Abstract.** In the list of challenges facing the world largest deltas, increased salt intrusion and its role in jeopardizing freshwater
supply is often ranked very high. Yet, detailed process-based studies of salt intrusion at the whole delta scale are limited and
the trends are regularly associated to global sea level rise. Here, using field measurements and a sophisticated 3D model that
integrates the riverine, rural, estuarine, and coastal dynamics within one numerical domain, we study salt intrusion at the scale
of the Mekong Delta in extensive detail. While many studies down-scale the salt intrusion problem to a topic within an estuary,
we show that the continental shelf is an intrinsic component of the delta, and its physical processes, such as monsoon-driven
ocean surge, directly influence salinity dynamics within the delta. Typical values of 20-40 cm surge over the continental shelf
contribute to up to 10 km of further salt intrusion. The delta's estuarine system is also more sensitive than many other systems
to upstream discharge variations. Furthermore, spring-neap variability plays a key role in salt intrusion in the delta. The
estuarine variability from a stratified to a mixed system between neap and spring tides develops 3D processes such as estuarine
circulation and tidal straining that become the main upstream salt transport mechanisms. The 3D nature of salinity dynamics,
and the role of upstream and downstream processes, suggests that compromising on dimension or extent of the numerical
domain, can limit the accuracy of predictions of salt intrusion in the delta. The study also showcases that riverbed incision in
response to anthropogenic sediment starvation in the last two decades, has increased stratification, and activated or magnified
3D salt transport sub-processes that amplify upstream salt transport. With all the external forces on the delta namely climate
change and altered hydrological regime by the upstream dams, due to deeper estuarine channels (driven by sand mining and
upstream impoundments), the delta itself is far more vulnerable to even mild natural events. This exemplifies the fundamental
importance of preserving the sediment budget and riverbed levels in protecting the world's deltas against salt intrusion.



## 1 Introduction

Over the past decade, Saline Water Intrusion (SWI) in the Vietnamese Mekong Delta (VMD) has been an issue of major concern and was identified as the key to land use and its future habitability (CGIAR Research Centers in Southeast Asia, 2016; Kantoush et al., 2017; UNDP, 2016). The 2016 and 2020 droughts resulted in significant disruptions (~450-650 k ha of crop loss) within the delta due to salt intrusion (IFRC, 2020; UNDP 2016). Eslami et al.(2019b) showed that there have been

increasing trends of salt intrusion and tidal amplification, even with increasing freshwater discharge from upstream (Eslami et al., 2019b; Li et al., 2017; Lu et al., 2014; Räsänen et al., 2017) that are being driven by bed level changes in response to anthropogenic sediment starvation. This sediment starvation is itself caused by sediment trapping by upstream dams (Fan et al., 2015; Koehnken, 2014; Kondolf et al., 2014, 2018; Kummu et al., 2010; Manh et al., 2015; MRC, 2011; Thi Ha et al., 2018) and intensive downstream sand mining (Brunier et al., 2014; Eslami et al., 2019b). Amid various reports of extreme salt

intrusion within the VMD, the present study shows the observed dynamics of salt intrusion in the dry season and applies an integrated state-of-the-art coastal-inland 3D numerical model to explain various physical processes contributing to the observed trends of increased SWI.

Previous studies on estuarine salinity dynamics can be roughly classified in three categories. 1) The tidally-averaged steady-state theories of competition between upstream dispersive versus downstream advective salt flux (Mac Cready and Geyer,

2010; Hansen and Rattray, 1965; Prandle, 2004; Pritchard, 1956; Savenije, 2005), 2) Non-stationary decomposition of salt transport processes such as Stokes transport, tidal pumping, tidal straining, gravitational circulation and residual flow (Banas et al., 2004; Díez-Minguito et al., 2013; Dyer, 1997; Fischer, 1976; Lewis and Lewis, 1983; Simpson et al., 2001; Uncles et al., 1985) and 3) 3D numerical modelling (Burchard and Hetland, 2010; Hetland and Geyer, 2004; Lerczak et al., 2009; Monismith et al., 2002; Simpson et al., 2001; Warner et al., 2005). The first category generally provides a primary framework

for studying long-term (seasonal or decadal) trends in estuarine systems. The second is an approach to understand the physical processes that dominate estuarine salinity dynamics. However, as estuarine systems are often quasi/un-steady and in transition, in absence of detailed high-resolution long-term measurements in these large bodies of water, any prognostic evaluation of estuarine dynamics resorts to numerical modelling (the third category of study). The competition between upstream dispersive and downstream advective salt flux, along with competition between buoyancy production and tidal mixing leads to variations

of salinity and its transport mechanisms in multiple temporal and spatial scales. To qualitatively describe the effect of freshwater inflow and tides in an estuarine system, a tidally averaged steady-state approach can be adopted. When quantitative in-depth knowledge is required, the spatio-temporal estuarine variabilities of spring-neap cycles, stratification, transversal variabilities, and their interactions become critical processes. To address these processes, 1D numerical models can perform adequately when an estuarine system is fully mixed; however, very often, stratification changes with spring-neap tidal cycles

(Jay, 2010; Jay and Dungan Smith, 1990) and together with morphological changes, can lead to estuarine salinity regime shifts (Geyer and MacCready, 2013; Simpson et al., 1990). These variabilities develop complexities that demand more detailed measurements and more sophisticated modelling approaches.



Given the gravity of SWI as a threat to the livelihood of the VMD, and other major deltas around the world, the available literature addressing it is limited. There are a number of field measurements (Gugliotta et al., 2017; Nguyen and Savenije, 2006; Nowacki et al., 2015) that provide snapshots of specific periods or regions, forming a qualitative impression of salinity in the delta. Modelling efforts are bound to analytical models (*e.g.,* Nguyen and Savenije, 2006) that are mainly focused on the steady-state salinity regime; and existing numerical modelling efforts are either in 1D (Smajgl et al., 2015; Tran Quoc Dat et al., 2011; Vu et al., 2018) with limitations in capturing various physical processes, or depth-averaged and variable in width (2DH) with limited domains (Tran Anh et al., 2018), both overlooking the effect of stratification. Furthermore, all these numerical studies are focused on projections in response to climate change, even though recent work has shown that channel changes are a much more significant control on barotropic and baroclinic dynamics (Eslami et al., 2019b; Vasilopoulos et al., 2020). Moreover, despite all the anecdotal reports of SWI in the VMD, it is only recently that a systematic analysis of available salinity data has shown how salinity is increasing within the VMD (Eslami et al., 2019b). The shear dimensions of the Mekong mega-Delta (~ 200 km x 200 km, with more than 8000 primary and secondary irrigation canals), the lack of detailed data, the computation capacity and limitations in the past numerical models have all conspired to make it difficult to study SWI on that scale and with the requisite temporal and spatial detail. However, recent developments in numerical modelling techniques now makes it feasible to integrate upstream and coastal processes within one numerical domain with different dimensions (Eslami et al., 2019a; Kernkamp et al., 2011; Martyr-Koller et al., 2017).

The aim of this study is to a) show the effect of subtidal variability (*e.g.*, surge, discharge and spring-neap tide) on SWI; b) address estuarine variability, and the 3D nature of SWI in the VMD and c) demonstrate how bed level changes in the VMD have changed the salt transport mechanisms, resulting in excessive SWI within the VMD. To that end, we measured salinity structure simultaneously along two distributary channels of the Hau River during a spring-neap tidal cycle. We previously (Eslami et al., 2019a) showed the importance of the VMD irrigation system on barotropic tidal propagation. Here, we use a state of the art high-resolution 3D numerical model (Kernkamp et al., 2011) of the Mekong estuarine system, including its system of primary and secondary channels and the continental shelf, incorporating a new 2018 bathymetry (Vasilopoulos et al., 2020) covering the principal distributary channel network. In a series of sensitivity analyses and perturbation simulations, we demonstrate the role of spring-neap variability, subtidal surge effect on SWI and show the temporal variation of advective and dispersive salt fluxes. Furthermore, we showed how SWI would vary during the 2016 drought, if an older bathymetry (2008) was still in place, thereby clearly demonstrating how recent (2008-2018) changes in bathymetry have dramatically altered salt transport mechanisms that exacerbate SWI in the Mekong Delta.

## 2 Environmental conditions and Methods

### 2.1 Area of study

The Mekong River (MR) springs from the Tibetan plateau in China, crosses Laos, Thailand and Cambodia until it forms the Vietnamese Mekong Delta (see Figure 1). The MR, carrying annually 350-500 G m$^3$ of water (Milliman and Fransworth,



2011), prior to entering the VMD, splits again into two branches, and forms the two systems of the Tien (Mekong) and Hau (Bassac) tidal rivers. Further downstream, the two rivers split into, in total, seven estuarine distributary channels (Figure 1b) with various channel width and depths (see Eslami et al., 2019a). Despite different properties, all these estuarine channels share similarities in their tidal dynamics, *i.e.*, they are all mesotidal hyposynchronous (Dyer, 1997; Nichols and Biggs, 1985), with strong spring-neap variations. Spring tides are dominated by semi-diurnal tidal species and neap tides are determined by

diurnal tidal components, introducing a sharp spring-neap variability. Morphologically, the lower ~80 km of the channels are tide-dominated and fluvially influenced, with widening and shallowing channels seaward, and significant mud content in the seabed (Gugliotta et al., 2017). Upstream of that until Phnom Penh, Cambodia, the channels are fluvially dominated and tidally influenced, with meandering channels that deepen seaward, with little mud content (Gugliotta et al., 2017). The VMD is characterized by strong monsoon-driven seasonal climate variations (Mekong River Commission, 2010), with the wet (July to

October, SW monsoon) and dry (December-May, NE monsoon) seasons contrasting dramatically. The wet season floods the floodplains of Cambodia and Vietnam, tidal propagation is damped by the large freshwater discharge from upstream and ocean saline water cannot penetrate more than a few kilometres into the estuarine channels (Eslami et al., 2019a; Gugliotta et al., 2017). However, during the dry season, tidal difference increases from ~ 1 m to 2 m in Can Tho (see Figure 1) and My Thuan (see Eslami et al., 2019a), salinity intrudes up to ~50-60 km  (assuming a 2 PSU threshold ) in the delta (Gugliotta et al., 2017;

Nguyen and Savenije, 2006; Nowacki et al., 2015), and the North-Eastern monsoon winds continuously create modest surge events (~10-30 cm) along the VMD coast (Eslami et al., 2019a). In this study, as we focus on SWI, we limit the timescale to the dry season, and specifically focus on the 2016 drought, driven by the 2015-2016 El Niño, and the subsequent excessive SWI within the VMD.

## 2.2 Field Campaign

Although drought is not new in the Mekong Delta, in 2016, there were numerous anecdotal reports of salinity reaching Can Tho City (*e.g.* UNDP, 2016). To measure the historical event, we aimed to measure maximum SWI during a high-water slack along the Hau River distributary network (Dinh An and Tran De), using moving boat measurement technique (Nguyen and Savenije, 2006; Savenije, 1986) . On two different dates, Apr-1st and April-9th of 2016, respectively during neap and spring tides, we measured salinity structures and intrusion lengths, simultaneously along the two channels. Starting from the estuary

mouth during the slack tide, we sailed upstream along the thalweg at approximately 30 km hr$^{-1}$ (*i.e.,* approximate tidal propagation speed) and at intervals of every 3 km measured salinity over depth (See Figure 1b for the salinity routes). Salinity was transformed to PSU from conductivity [μS cm$^{-1}$] and temperature [°C] measurements made by of the conductivity sensors (SensorNet TetraCon 700 IQ).

## 2.3 Model description

Delft3D-Flexible Mesh (Delft3D-FM) (Kernkamp et al., 2011) is a state of the art, open-source numerical model that solves the incompressible Shallow Water Equations in an orthogonal unstructured staggered grid discretization (Kramer and Stelling,



2008). The model applies a finite volume conservation formulation of mass, momentum, turbulence, salinity, temperature, and conservative matter. The barotropic pressure coupling is implicit, but baroclinic pressure is treated explicitly. spatially applying a finite volume staggered-grid discretization. The model implicitly solves conservation of mass, momentum, transport of matter

and the Eckart (Eckart, 1958) or UNSECO (UNESCO, 1981) equation of state in three dimensions, for water level, horizontal and vertical velocity and salinity concentration. In the vertical, a so-called z-layer (fixed) scheme is applied, along which hydrostatic pressure assumption, and a Manning bottom friction treatment is adopted. Furthermore, in the vertical a $k - \epsilon$ turbulence closure is applied (Rodi, 1993). Vertical advection of momentum and the turbulence properties are solved with a first order upwind scheme, but vertical transport of matter is treated with a higher order discretization scheme. The reader is

referred to Kernkamp et al., (2011) and Martyr-koller et al., (2017) for more details on Delft3D-FM development and background.

**2.4 Model set-up and modelling approach**

The previously developed coupled 1D-2DH barotropic model of the VMD  (Eslami et al., 2019a) was further extended and modified in this study to assess detailed salt intrusion processes. The 2DH parts of the network were expanded to 3D, and the

1D channels were expanded in depth to 2DV (width-integrated but depth-varying), following identical vertical layering as the rest of the model. Furthermore, the Western continental shelf was added to the model domain and downstream of Can Tho and My Thuan (see Figure 1), the model domain was updated to 3D (previously in 1D). The horizontal model resolution was increased to approximately 200 m along the channels and 50-200 m across the channels (depending on channel width). The vertical resolution of the z-layers was set to 0.5 m, *i.e.*, 20 vertical layers in 10 m of depth. Vertical resolution was gradually

increased below 15 m depth to save computation time in the deeper continental shelf. Following Eslami et al. (2019a), the re-calibrated spatially-varying Manning bed friction coefficient was lower along the downstream estuarine distributary channels (0.02 s m$^{-1/3}$) and increased to 0.04 s m$^{-1/3}$ along the upstream rivers.

From upstream, the model was forced with the daily averaged discharge at Kratie, Cambodia (nearly 500 km upstream), and from downstream by 13 leading astronomic tidal components located 70 km offshore. Measured subtidal WL at Binh Dai (see

Figure 4) was super-imposed to the offshore boundary with its associated time lag (similar to Eslami et al., 2019a) to replicate subtidal water level variations, representing the effect of ocean surge at the boundary. The negligible (very low in simulation year of 2016) daily averaged discharge from the Tonle Sap Lake was calculated as the difference between the discharge entering the VMD (as observed for the Tien and Hau Rivers at Tan Chau and Chau Doc) and the Mekong River discharge as observed at Kratie (considering the time lag between Kratie and the VMD). The Neuman conditions were assigned to the

cross-coastline boundaries and offshore wind is considered uniformly distributed over the model domain. Evaporation gradually increased from 4 to 7 mm/day from January to April (Kumiko et al., 2008; Kummu et al., 2014), transpiration was neglected, and precipitation was assumed zero during the dry season. Spatially varying water demand was incorporated in a manner identical to Eslami et al., (2019a) and a total of 52 sluice gates (Figure 1), contained the irrigation system from SWI and remained closed throughout the simulations. **Table 1** provides a summary of the input data and their sources.





As salt intrusion is a dry season phenomenon, the simulations ran from January to end of April, and the first month of simulation
        was considered as spin up time. In order to examine various physical processes, following a method of factors separation
        (Buschman et al., 2010; Eslami et al., 2019a; Sassi et al., 2011), various model forcings were specifically turned on and off in
        order to evaluate their effect and importance on SWI. Riverbed levels have significantly changed in the past 20 years (Eslami
        et al., 2019b; Vasilopoulos et al., 2020). In order to demonstrate the role of riverbed level changes on SWI, the calibrated
model with the 2018 bathymetry was also run with an older 2008 bathymetry. The 2008 bathymetry was partly obtained from
        Mekong River Commission (MRC) and partly obtained from the Southern Institute for Water Resources Planning (SIWRP),
        while for 2016 simulations the 2018 bathymetry of  Vasilopoulos et al. (2020) was used.

## 3 Results

### 3.1 Field campaign

Amid various reports of historical SWI up to 90 km (near Can Tho City) in 2016 (e.g., UNDP, 2016), our field campaign failed
        to measure values that substantially exceeded previously published observations. Along the Dinh An channel, we measured
        maximum ~50 km of salt intrusion (with 2 PSU as the threshold) during spring tide (see Figure 2), similar to observations of
        Nguyen and Savenije (2006). This triggered various questions on the timing and the extent of SWI as the drought reports were
        still in the press. Despite not measuring the historical salt intrusion length, the observed trends were intriguing. Within the
Dinh An Channel, SWI length was slightly higher during the spring tide, but salinity near the mouth was significantly higher
        during the neap tide. Stratification reduced remarkably from neap tide to spring tide in the Dinh An channel, but did not change
        significantly in the Tran De channel. Figure 3 shows various environmental parameters during the 2016 dry season. Figure 3a
        shows the observed daily averaged wind speed and direction (pre-dominantly NE monsoon wind) along with tidal and the
        Godin-filtered (Godin, 1972) subtidal water level at Can Tho, where we see the subtidal response of water level to wind (also
see Eslami et al., 2019a). Figure 3b shows subtidal discharge at Can Tho, which is controlled by upstream freshwater supply
        and downstream subtidal variations (Eslami et al., 2019a).
        Stationary, 2-hourly measurements of depth-averaged salinity (see section 3.2 for further detail) at two stations, Dai Ngai and
        Cau Quan, showed that salinity was highest around the second weeks of February (period-1) and March (period-2) and declined
        significantly past mid-March, before the field campaign took place. These two periods do follow episodes of lower subtidal
discharge, but do not necessarily coincide with the lowest freshwater discharge periods (*e.g.*, see Mar. 22$^{nd}$ to Apr. 1$^{st}$). In
        terms of tidal stage, both periods coincide with the transition period of neap to spring tide. Both periods of high salinity also
        coincide with ocean surge events, although period-1 correlates with a significantly stronger surge. Overall, this implies that a
        combination of physical forces determines salinity level in the delta and cannot be reduced to simply upstream discharge or
        downstream tidal forcing. At a sea boundary it is not only the tidal variation, but also the residual current intensity and river
discharge in the preceding period that determine the overall salinity at the ocean boundary. These complexities demand an



integrated 3-dimensional modelling, with as many environmental forcings as possible to explain the barotropic, baroclinic, intra-/subtidal processes that drive SWI within the system.

## 3.2 Model validation

For the year 2016, a series of model performance statistics Pearson correlation coefficient ($R^2$) and Nash-Sutcliffe coefficient of efficiency NSE (Nash and Sutcliffe, 1970) were calculated to examine the Delft3D-FM model performance (see Figure 4a, b, c, i & j). Comparison of stationary salinity observations with the model was slightly more complicated. The observations are at 2-hourly intervals, with irregular gaps in between. The actual measurements are carried out manually, at a point off the riverbank, which means that it is never a fixed point. Manual measurements over depth are then averaged by $S = (S_{20\% \ depth} + 2 \times S_{50\% \ depth} + S_{80\% \ depth})/4$, where $S$ is the presented observed depth-averaged salinity (personal communication with SRHMC and SIWRP). Furthermore, it will be shown later that salinity within the VMD is highly sensitive to small variations of freshwater supply and other subtidal forcings. Therefore, although the assumed water demand distribution is as close as possible to reality (see also Eslami et al., 2019a), in practice, it is subject to spatial and temporal variations, especially in specific drought years, that are not input to the model. This makes the model-observation comparison a delicate matter.

By calculating a depth-averaged salinity in the same fashion, the qualitative comparison was made for intra- and subtidal variations at multiple stations (see Figure 4d, e, f, g, k & l). Subtidal salinity is modelled reasonably well in all stations, but intra-tidal salinity at Dai Ngai (Figure 4-l) is underestimated. However, given the low salinity at Dai Ngai and its location near a major primary water distribution canal, as well as multiple islands in the vicinity, and in proximity of an estuarine junction, comparisons at this location are difficult. Furthermore, Figure 5 shows a comparison of the modelled and observed (field campaign) HWS estuarine salinity structures along the two distributary channels of the Hau River, during neap and spring tide. The Dinh An channel clearly transitions from a stratified estuary during neap tide to a partially-mixed system during the spring tide, while the Tran De channel remains a moderately stratified to partially-mixed system over the entire period. In a qualitative comparison, these patterns were also reproduced by the model, although the model tends to have slightly smaller top-bottom differences in salinity and smaller SWI than observed. The model-data comparison shows that the model is able to represent the spatial and temporal patterns in SWI reasonably well and can develop the estuarine spring-neap variability.

## 3.3 General trends of SWI

In the previous section, we showed a strong spring-neap variability along the two lower distributary channels of the Hau River. To study SWI along all the estuarine distributary channels, Figure 6 shows the temporal variations of subtidal (tidally-averaged) depth-averaged SWI length (with 2 PSU threshold), along seven distributary channels, as well as the over-depth salinity difference, indicating stratification. Overall, the VMD experiences variable SWI of ~30-70 km during the dry season. Despite differences in the magnitude, with the exception of the lower My Tho estuarine distributaries (Dai and Tieu), the lower Hau and Tien River estuarine distributary channels follow similar trends of SWI. The estuarine systems of Tran De – Dinh An





and Co Chien - Cung Hau, as well as Ham Luong channel show analogous temporal variations of SWI length. Furthermore, in terms of stratification, Co Chien – Cung Hau and Tran De – Dinh An show similar bi-weekly spring-neap variations, *i.e.*, during the neap tide, Dinh An and Co Chien (the two deeper channels), experience much higher stratification than Tran De and Cung Hau. Between the lower distributary channels of Cho Lach, stratification patterns of the single Ham Luong channel also resemble those of the deeper Dinh An and Co Chien channels.

However, within the lower My Tho channels, the narrower and shallower Dai and Tieu have more continuous SWI patterns. This can be partially explained by the limited temporal estuarine variability in terms of stratification. While all channels show increased stratification during neap tide, this is limited in Dai and Tieu. These estuarine channels remain partially to well-mixed throughout the dry season, and consequently, show a slower response to short-term environmental forcing anomalies, *i.e.*, they mainly follow the large-scale trends of upstream discharge variations. In addition, as the Mekong River discharge gradually increases in April, so does water demand within the provinces along the Tien River. As their received freshwater supply during the dry season is in total ~5% of the total VMD freshwater inflow, upon increasing salinity in the channels, the minor spatial variations in discharge and downstream water level gradient are not sufficient to compensate for the upstream baroclinic pressure gradient. Therefore, they take longer in the dry season, towards the wet season, to become fresh again.

Since the model was able to reproduce the intra- and sub-tidal processes, as listed in Table 2, in a number of simulations, we examined the role of various processes in determining SWI. As the measurement campaign was carried out in the lower Hau River and its two distributary channels and most of the estuarine channels seem to follow similar SWI trends in time, for further detailed analysis we focus only on the Hau River estuarine channel network.

## 3.4 Ocean surge effect

Figure 7a shows modelled tidal and subtidal water levels at Can Tho, including (reference simulation called Ref.) and excluding (referred to as NS) offshore ocean surge, showing how higher subtidal water levels are associated with ocean surge and correlated to strong NE monsoon winds. Figure 7b shows the effect of surge on subtidal discharge in Can Tho, upstream of Dinh An – Tran De estuarine network. Note that even in the absence of ocean surge (NS simulation), there is still a strong spring-neap signal in subtidal discharge. During the spring tide, Stokes transport generates an upstream water level gradient that releases discharge towards neap tide (also see Eslami et al., 2019a). Figure 7c & d show the simulated change in salinity in absence of ocean surge near two stationary observation points, and Figure 7e shows salt intrusion length in the Dinh An channel with and without the effect of ocean surge. In general, increase in water level driven by ocean surge, can increase subtidal stationary salinity by 20-30%, 25 km from the river mouth, and increase subtidal salt intrusion length by as much as 10 km (see Figure 7e). This underscores the interconnectivity of baroclinicity within the Mekong estuarine system with processes that develop hundreds of kilometres offshore. Therefore, depending on the sensitivity of the application, accurate forecasting of SWI demands a system that at least incorporates complete barotropic variations of the continental shelf.





### 3.5 Spring neap cycle

In section 3.3, we briefly touched upon the spring-neap variability of the estuarine channels and its potential role on SWI.
Here, by limiting the offshore tidal constituents to a single semi-diurnal $M_2$ tide (with amplitude $M_2+ 0.5 S_2$, see Table 2),
while other environmental forces remain unchanged (*e.g.*, ocean surge, discharge variability, water demand etc.), we removed
the spring-neap cycle from the estuarine system forcing, which resulted in up to ~30 km lower SWI length (Figure 7e).
Contrarily, by limiting the tidal species to a diurnal constituent (with amplitude $K_1+O_1$), effectively reducing tidal mixing,
salinity intrusion length increased up to ~20 km. By limiting the tidal constituents to $M_2$ and $S_2$ to generate a spring-neap cycle
($M_2S_2$), in the absence of 11 other tidal constituents, the modelled salt intrusion length was still generally close to the reference
simulation, accentuating the significant role of spring-neap variability on SWI. Figure 8 and Figure 9 show estuarine variability
in the Dinh An channel over two periods: period-1) high SWI in early February (Figure 8) and period-2) prior to the field
campaign (Figure 9). Period-1, during its first 10 days, is marked by a strong ocean surge that amplified channel depth, hence
stratification and period-2 generally shows lower salinity that is partly driven by lower salt content due to higher discharge
along the second half of March. Nonetheless, both periods show that periods of stratified salinity structure are followed by
periods of high upstream salt flux (see section 3.6). Maximum salinity is reached during the transition of the system from
stratified (neap) to [partially] mixed (spring) states when the system responds to tidal forcing. When the system is not stratified
anymore, average salinity starts declining with advective transport dominating total salt flux in the downstream direction.
Given the similar temporal trends of estuarine variability in different estuarine channels of the VMD, this indicates how
important the role of spring-neap variability is in increasing total salt intrusion in most of the VMD. By decomposing the salt
fluxes in section 3.6 we will further expand on the role of tidal variability and the effect of spring-neap cycle, along with other
environmental forces on SWI in different periods.

### 3.6 Salt fluxes

To better understand how the temporal changes in forcing (spring-neap, discharge variation, ocean surge) or bathymetry cause
changes in the SWI, we studied the subtidal salt transport and decomposed it into its different contributions. Following (Banas
et al., 2004; Dyer, 1997; Lewis and Lewis, 1983; Simpson et al., 2001; Uncles et al., 1985) and borrowing the notation of
Díez-Minguito et al., (2013) notation, total over-depth salt flux is:

$$f = \int_{-h_0}^{\eta} u\, s\, dz \; ; \qquad h = h_0 + \eta, \qquad\qquad\qquad (1)$$

Where $u$ is the along channel velocity per unit of width, $s$ is the salt content, $h_0$ is the mean depth, $\eta$ elevation and $h$ total
depth. Velocity and salinity can be written as:

$$u = \underline{u} + u_v; \quad s = \underline{s} + s_v; \quad u = \overline{u} + \tilde{u}; \; s = \overline{s} + \tilde{s} \qquad\qquad\qquad (2)$$

With __ denoting the depth-averaged, ‾ referring to tidally averaged (subtidal), $_v$ showing deviation from depth-averaged
and ~ deviation from the subtidal signal. As the flux calculation is carried out on numerical model results and over a given



cross-section, the integration over depth and width are carried out within the model vertical and horizontal model resolution.

As velocity is considered constant within a numerical grid, within the horizontal plain, instead of $u$, its product with vertical grid area $a$ is used, which outputs the along channel discharge through a grid cell $q$:

$$q = u \times a, \tag{3}$$

Therefore, per cross-channel grid cell, the salt flux is calculated over depth. Total salt flux can be decomposed to eight components:

$$\overline{f} = \underbrace{\overline{h}\,\overline{q}\,\overline{s}}_{T_1} + \underbrace{\overline{s}\,\overline{\tilde{\eta}\,\tilde{q}}}_{T_2} + \underbrace{\overline{q}\,\overline{\tilde{\eta}\,\tilde{s}}}_{T_3} + \underbrace{\overline{h}\,\overline{\tilde{q}\,\tilde{s}}}_{T_4} + \underbrace{\overline{\tilde{\eta}\,\tilde{q}\,\tilde{s}}}_{T_5} + \underbrace{\overline{h}\,\overline{q_v\,s_v}}_{T_6} + \underbrace{\overline{h}\,\overline{q_v\,s_v}}_{T_7} + \underbrace{\overline{\tilde{\eta}\,\overline{q_v\,s_v}}}_{T_8}, \tag{4}$$

Where $T_1$ is the Eulerian non-tidal flow, summing up salt flux by river discharge, Stokes return flow and all influences on the mean flow. $T_2$ is associated with Stokes transport, $T_3$, $T_4$ and $T_5$ are tidal pumping terms correlating tidal variations of depth and depth averaged current and salinity. $T_6$ is a term associated with covariance of depth-varying current and salinity, namely, tidal straining. $T_7$ refers to estuarine circulation, consisting of vertical gravitational circulation and other vertically sheared

flows (e.g., strain-induced periodic stratification) and $T_8$ is a normally negligible triple correlation of water level variation and depth-varying deviation of velocity and salinity. Note that above fluxes are over depth and per cross-channel grid cell. These fluxes are then integrated over the numerical domain across the cross-section, in cross-channel horizontal resolution.

Figure 7f & g show different cross-sectionally integrated salt fluxes in the two lower distributary channels of the Hau River, nearly 15 km from the sea. In both channels, the Stokes transport is always positive (upstream), and the residual Eulerian

transport is always negative (downstream). The Dinh An channel, because of its larger depth and cross-sectional area (nearly double in width) shows salt fluxes larger by nearly two fold. In both channels, the Stokes transport peaks by spring tide, driven by increased amplitude of semi-diurnal tide and its associated mixing, while total salt transport is mainly downstream, driven by advective Eulerian transport. Total salt flux in the Dinh-An channel can be characterized as more event-like compared to the Tran De channel. This is especially driven by its spring-neap variability in stratification due to the larger depth of the Dinh

An channel. In both channels, estuarine circulation increases and dominates salt flux with lower reduced mixing during neap tide. Prior to the two periods with maximum salt intrusion (early February and March) upstream salt transport peaks especially in the Dinh An channel. These peaks start with peaks in estuarine circulation during neap tide and sustain with peaks in tidal straining in the onset of transition to the spring tide. While the terms associated with tidal pumping show limited contribution to the total salt flux, tidal straining becomes an important term during peak salt intrusion events. Tidal straining maximizes in

transition between neap and spring, when semi-diurnal tidal amplitude increases, but the system is yet to transition from stratified to partially mixed. The assessment of salt fluxes further visualizes the importance of various forcings as well as geometry/bathymetry, i.e., channel depth in SWI. Therefore, by leveraging a similar analysis, we will further investigate what salt transport mechanisms are influenced when riverbed levels change within the estuarine system of the VMD.



### 3.7 Effect of bathymetry

The calibrated model was run with the 2008 bathymetry to demonstrate how bed level changes have influenced salt intrusion in the VMD. It was previously shown that the Ham Luong estuarine channel (see Figure 1) has experienced the largest increase in salt intrusion compared to other estuarine channels (Eslami et al., 2019b). That study also showed that tidal amplitude in the Tien river system has increased by nearly 40%. Therefore, in this section, we focus on the model results of the Ham Luong estuarine channel. Comparison of the 2008 and 2018 (Vasilopoulos et al., 2020) bathymetries shows an average ~1.5 m incision

in Ham Luong channel during this period, but this value increases to ~3 m along the thalweg, which implies significant influence on tidal propagation and fresh-saline water dynamics. **Figure 10**a shows that under identical environmental forces, salt intrusion length would be 5-10 km shorter, had the riverbed levels not changed in the past decade. **Figure 10**b & c show various salt fluxes at two cross-sections, 20 km and 50 km from the sea, respectively. In terms of significance of various fluxes, we observe similar trends between Ham Luong and Dinh An channels (see also Figure 6 and Figure 7). Total salt transport,

especially during neap tide when the system is more stratified, has increased by nearly 50%. This is mainly driven by fluxes associated with gravitational circulation and tidal straining, directly correlated with increased stratification in response to deeper riverbed levels. Upstream Stokes transport that is mainly driven by increased semi-diurnal tidal amplitude and the cross-sectional area has also increased by nearly 15% under the more recent bathymetry. However, as Stokes transport is maximum during spring tide when stratification is minimized, its effect is compensated by the advective downstream residual

transport, dominated by river discharge and Stokes return flow.

    This exercise shows that increased river depth directly and profoundly impacts salt flux mechanisms. For example, 50 km from the estuarine mouth (**Figure 10**c), there would nearly be no salt flux due to estuarine circulation. However, under the current (2018) bathymetry, salt transport by estuarine circulation is the dominant mechanism, and by tidal straining has nearly doubled. Note that the most impacted processes are highly three dimensional with temporal variations in their significance. Salt transport

by estuarine circulation is maximized during the neap tide, and transport by tidal straining is maximized in the early transition of neap tide to spring tide, while Stokes transport that is often treated reasonably well, even in 1D numerical/analytical approaches, peaks during spring tide. These dynamics suggest that not only the bathymetry change has shifted the salt flux regime of the estuary, but it has also developed to a more complicated 3D system that requires a more sophisticated treatment. Therefore, it is plausible to argue that one-dimensional treatment of such a complicated system, as it is common practice,

cannot accurately reflect changes in the salinity dynamics of the Mekong Delta.

### 4 Discussion

    Previous modelling studies of salinity in the Mekong Delta can be classified into three categories: 1) applying analytical stationary models along with field measurements (Nguyen et al., 2008; Nguyen and Savenije, 2006), with temporal and spatial limitations; 2) 1D numerical studies (Khang et al., 2008; Smajgl et al., 2015) that include the system of primary and secondary

channels but exclude the continental shelf; these models need to force downstream salt concentration boundary conditions and



due to the depth-averaged nature, are limited in physical representation; or 3) offline numerical coupling of 1D and 2D (Tran Anh et al., 2018) or 3D (Nguyen and Tanaka, 2007), limited to lower Hau River. Given the most recent numerical developments and computational power, for the first time, it was possible to efficiently model the entire Mekong Delta in a combination of 2DV and 3D. This provided a powerful tool with a fully integrated online-coupled model of upstream river

sections, irrigation canals, estuarine channels, and the continental shelf. With this tool, 3D processes are fully represented, and salt transport mechanisms were naturally [numerically] modelled. Many 1D modelling exercise have to resort on forced or analytical definitions of downstream salinity as they do not extend offshore. As salinity at the estuary mouth is highly variable with principal effect on SWI in the estuary, here, we do not compromise on artificially forced boundary conditions and can fully capture the interconnectivity of coastal-inland salinity dynamics in response to various forcing's.

**4.1 Three-dimensionality of SWI**

The combination of variable external forcing in discharge, tide, wind etc., as well as large variations in geometry and bathymetry, and the fresh-saline water dynamics result in a wide range of estuarine characteristics around the globe (Dyer, 1997; Fischer, 1976; Geyer and MacCready, 2013; Valle-Levinson, 2010). These characteristics are projected in salt transport mechanisms that lead to the realized salt balance or saline water intrusion in these systems. These mechanisms can be more

depth-averaged (2D) when buoyancy is limited or depth-varying (3D) when the system is more stratified. Examples of systems with more depth-averaged processes are Conwy Estuary, where salt transport is dominated by tidal pumping (Simpson et al., 2001), or the Guadalquivir Estuary where nontidal and Stokes transport mechanisms lead tidal straining (a 3D process) as most important salt fluxes (Díez-Minguito et al., 2013). On the other hand, in a system such as the Columbia River Estuary salt transport is more dominated by quarter-diurnal signal (Mac Cready and Geyer, 2010) of tidal straining (Burchard and Hetland,

2010; Jay and Musiak, 1994; Jay and Smith, 1990a, 1990b), which is a three-dimensional process.

This study, for the first time, provides a coherent and integrated picture of salinity dynamics in the Mekong Delta and shows SWI within the delta has a 3D character with large variability in different temporal and spatial scales. Previously, estuarine salinity of the VMD was either measured stationary (Eslami et al., 2019b; Nowacki et al., 2015) or in along-channel snapshots during spring tide (Nguyen and Savenije, 2006). The estuarine salinity structure and its neap-spring variability was studied

here for the first time. Estuarine stratification variability can have significant implications for 3D estuarine circulation (Jay and Smith, 1990a&b; Jay, 2010;). We showed that temporal changes in mixing, driven by the spring-neap cycle have dominant role in SWI within the VMD. The gravitational circulation and other vertically sheared flows resulting from depth-varying forcing dominate salt transport during the neap tide, and tidal straining becomes the significant process in transition to the spring tide, both of which are highly 3D processes. During the spring tide, the partially-to-fully mixed system acts more as a

depth-averaged system, and generally flushes salt with non-tidal residual flow, predominantly a combination of river discharge and the return flow of the strong nonlinear Stokes transport. In short, we can say the system imports salt in a 3D fashion but flushes in a depth-averaged manner.





## 4.2 Subtidal variability

It was previously shown that under low discharge regimes, ocean surge can travel hundreds of kilometers upstream (Henrie
and Valle-Levinson, 2014) and influence discharge division in multi-channel estuarine systems (Eslami et al., 2019a), but its
effect on salt intrusion was not addressed before. This study quantifies the effect of ocean surge on salt intrusion (up to 20%
stationary or 10 km in intrusion length of subtidal depth-averaged salinity), along with other subtidal variations such as
discharge and spring-neap variability. The salinity peaks of the VMD during the 2016 drought, not only coincided with low
discharge periods, but also to high surge events. While discharge variation inversely changes salt intrusion in a relatively well-
understood manner (Abood, 1974; Gong and Shen, 2011a; Monismith et al., 2002; Ralston et al., 2010), we could conclude
from the sensitivity analysis that a positive surge increases salt intrusion , but it was not directly clear from the sensitivity
analyses with real forcing  (e.g., Figure 7) how exactly ocean surge impacts salt intrusion. To examine system response in time
and space, a perturbation analysis with synthetic forcing was set up. A series of perturbation simulations (for the entire Mekong
model) were run for 160 days with constant discharge (3000-6000 $m^3s^{-1}$) and different tidal forcings (only $M_2$, only $K_1$ or
$M_2S_2$) to arrive at a stationary/equilibrium state. These stationary states were then interrupted with perturbation of upstream
discharge (in Kratie) or subtidal offshore water level. A positive/negative peak/trough of a 10-day discharge/surge wave was
forced at the upstream/downstream boundary and the adaptation time and the salinity intrusion response were examined. Table
3 summarizes the list of 39 perturbation simulations.

Figure 11 summarizes the results of the perturbation analysis for the Dinh An channel, while similar trends could also be
observed in other channels, except lower My Tho distributary channels (see section 3.3). Generally, the system with diurnal
tide (less mixing) experiences longer SWI length than a system with only semi-diurnal tide (more mixing), and the difference
increases (from ~12 km to ~18 km) with decreasing discharge from 6 to 3 k $m^3$ $s^{-1}$ (Figure 11a). Furthermore, as spring-neap
variability increases maximum salt intrusion compared to when the system is forced by only diurnal or semi-diurnal tide, the
variations and sensitivity to spring-neap cycle increases non-linearly with lower discharge and longer SWI length. Figure 11b
& c show the system response to upstream discharge (in Kratie) variation and Figure 11d & e show salt intrusion adaptation
to subtidal changes of coastal water levels. Both system responses are examined for conditions with only diurnal (b & d) or
semi-diurnal tides (c & e). There is an immediate reaction to subtidal water level changes but the effect of discharge in Kratie
is delayed nearly 2-3 days until the upstream wave reaches the delta. In response to subtidal variations (both upstream discharge
or downstream surge), a semi-diurnal system (more mixing) adapts faster and returns quicker to stationary conditions than a
diurnal system (less mixing). In general, the longer the salt intrusion, the slower it is to go back to its unperturbed conditions.

Estuarine response time to river discharge variation has been addressed by relating SWI length to a power law of river discharge
$L = \alpha\, Q_f^n$; with $Q_f$ as discharge, and $\alpha$ and $n$ constants (Abood, 1974; Gong and Shen, 2011b; Monismith et al., 2002; Ralston
et al., 2010). The lower Hau River, in terms of response to discharge showed similar behavior as Madaoman Estuary (one of



the estuaries of the Pearl River Delta), scaling with $n \approx -0.5$. This value is smaller than those associated for estuaries such as Hudson River (-0.33), Skagit (-0.25) and Merrimack (-0.19) implying a more sensitive system to discharge variations. Furthermore, similar to previous observations (Gong and Shen, 2011b; Hetland and Geyer, 2004), the system has a slightly shorter response time to an increasing discharge (reducing SWI) than a decreasing discharge (increasing SWI). Finally, the net
increase in salt intrusion length due to a discharge trough is stronger than the net decrease due to a discharge peak.

The surge effect on SWI introduces a slightly different behavior in terms of estuarine response, which was not addressed previously. Generally, the adaptation time (17-20 days to a 10-day surge) is shorter under semi-diurnal (mixed) tidal regime than under diurnal (stratified) tidal forcing. Similar to discharge variation, under a diurnal (stratified) tidal forcing, up-estuary
dispersive salt flux (by rising surge) is slower than down-estuary advective flux. This means that the system reacts slightly faster to the falling surge than a rising surge. However, as downs-estuary advective flux is stronger under a semi-diurnal tidal regime, adaptation time to a rising surge is slightly shorter than to a falling surge. As the SWI response to upstream discharge variation is delayed and peak-like, the response to downstream surge is immediate, but longer as it entails a secondary response in discharge in the second limp of the surge. This is interesting, because a subtidal water level trough can also temporarily
increase salt intrusion after it reduces that in the first response. Therefore, the system adaptation time to the surge is longer (up to 20 days). This is elaborated further in Figure 12, showing subtidal water level and discharge response to surge and discharge perturbation, upstream of the Dinh An channel in Can Tho. The rising upstream discharge translates to a simple rise in water level and discharge in Can Tho. However, a rising surge, as it increases water level in Can Tho, it decreases discharge, and this discharge blockage is followed and compensated by a rising discharge. The rising surge increases salt intrusion by
increasing channel depth and blocking upstream discharge, but due to its rebound effect on discharge in the falling limb, a rise in salt intrusion is followed by a decline. This implies that the continental shelf, with its physical processes, is an intrinsic component of the very same delta system influencing salt intrusion in the delta. Therefore, 1D models that exclude this crucial part of the system (*e.g.,* Smajgl et al., 2015; Tran Quoc Dat et al., 2011; Vu et al., 2018), can be prone to significant errors.

**4.3 Impact of riverbed level changes**

Examples of increased salt intrusion in response to riverbed level incision has been shown in other estuaries such as, amongst others, Hudson, Eems, Elbe, Loire, Schelde and Tanshui (Liu et al., 2004; Ralston and Geyer, 2019; Winterwerp and Wang, 2013). Had the riverbeds within the VMD remained at their 2008 elevations, the impact of salt intrusion during the 2016 drought would have been significantly lower. The deeper channels activate/magnify 3D sub-processes of upstream salt transport that can increasingly amplify salt intrusion. The VMD and many of its distributaries, occupy a wider area compared
to *e.g.*, Hudson River, within the estuarine parameter space (Geyer and MacCready, 2013). Because of the spring-neap variability and discharge sensitivity to external forces, the wide range of possible mixing parameters, ranging between 0.6-2, shifts the VMD between areas occupied by the Hudson River to the Columbia and Conwy River, etc. Unlike the Hudson River Estuary, where increased salt intrusion could not be associated to gravitational circulation (Ralston and Geyer, 2019), here we



quantify clear increase in gravitational circulation, as well as tidal straining, related to increased stratification. This implies
that perhaps gravitational circulation and tidal straining tend to be more sensitive to depth variation in systems with higher
spring-neap variability, compared to systems that tend to remain mixed or stratified in time. As the main driver of increased
SWI Within the Ham Luong channel (as well as others), approximately 10 km additional salt intrusion can directly be
associated with riverbed level incision. Although changes in SWI is often associated to sea level rise and global climate change,
this detailed study of salinity in an Asian mega-delta emphasizes that riverbed level changes, have been the key determinant
of changes in salt intrusion in the VMD (also see Eslami et al., 2019b).

## 5 Conclusions

This study has provided new understanding into the physical processes that dominate salt intrusion in the Mekong Delta. This
was done by carrying out field measurements as well as applying a novel 2DV-3D numerical modelling of the entire Mekong
Delta and its continental shelf that could reproduce various observed salinity patterns. This facilitated efficient integrated
modelling of the continental shelf, estuarine channels, and the complicated irrigation system. Amid the 2016 drought and
extreme salt intrusion within the VMD, we measured estuarine salinity structure along two lower distributary channels of the
Hau River. We observed significant estuarine variability between neap (strongly stratified) and spring (partially- to well-
mixed) tides. It was shown that most of the estuarine channels of the VMD follow the strong spring-neap variability, except
the lower My Tho distributary channels that demonstrate a more mixed system. We quantified that increased stratification
during the neap tide leads to estuarine circulation dominating upstream salt transport during the neap tide. In transition to
spring tide, tidal straining stimulates upstream salt flux but as we reach the spring tide, with increased mixing (reduced
stratification), down-estuary advective salt flux dominates and flushes salinity. Studying salt fluxes showed that salt transport
in the system is highly a 3D phenomenon.

Moreover, the system is significantly sensitive to external forces. This study, for the first time, showed that subtidal ocean
surge can impact the temporal variation of salinity by up to 20%. During dry season, typical variations (20-40 cm) in surge
showed larger (5-10 km) impact on salt intrusion than typical variations (1000-2000 $m^3$ $s^{-1}$) in upstream discharge (2-5 km).
The Mekong estuarine system demonstrated a short (a few days) flushing and transition times in response to subtidal variations
(*e.g.*, ocean surge or discharge variations). A discharge pulse at Kratie (Cambodia) travels to the delta in 3-4 days and its effect
lasts 3-4 days longer than the pulse itself. The ocean surge, however, leads to a near-immediate response in salt intrusion and
because of its rebound effect on discharge, its impact can last twice longer than the surge period itself.

The complexity and three-dimensionality of salt transport mechanisms in this deltaic system demands a 3D model to capture
the delicate temporal variations of upstream salt transport. The sensitivity of salinity on ocean-estuary interaction at the mouth,
with the continental shelf acting as an intrinsic part of the estuarine system demands a model that extends far offshore to
integrate both barotropic and baroclinic interactions of the inland-coastal system. Therefore, although 1D numerical models





can be useful for studying general trend dynamics, to analyse or forecast salinity within the delta, using a 3D integrated (inland-coastal) numerical domain that incorporates all processes is inevitable.

This study also emphasizes on the importance of riverbed levels in controlling salt intrusion. By modelling salt intrusion during the dry season of 2016 with the recent (2018) and old (2008) bathymetries, we showed the significance of riverbed level changes on 3D processes of SWI within the VMD. Although upstream dams, by altering the hydrological cycle, do have

significant impact on freshwater supply to the Mekong Delta, the delta itself is far more vulnerable to external forces (*e.g.*, drought) due to its deeper channels, driven by sediment deprivation from upstream dams, and shear amounts of sand mining within the delta. This is substantial further evidence that suggests riverbed levels are existential assets to preserve livelihood and the way of life within the Mekong Delta.

## 6 Data availability

DFlow-FM is an opensource numerical model https://oss.deltares.nl/web/delft3dfm. The underlying gauge data (observed water level, discharge, and salinity) provided by the SIWRP, following the organizational policy, can be provided upon request for non-commercial use. The wind data originally developed by NCEP/NOAA, can be downloaded from the DHI repository unde https://www.metocean-on-demand.com/#/main. The geometry data of the network of primary and secondary channels can only be provided in direct communication with the SIWRP. The 2008 bathymetry data is partly received from the Mekong

River Commission https://portal.mrcmekong.org/data-catalogue and partly received from the SIWRP and the 2018 contemporary bathymetry data can be downloaded from https://hydra.hull.ac.uk/resources/hull:17952. Nevertheless, all the data and models that cannot directly be accessed by the public can be provided to the reviewers for any validation or reproduction.

## 7 Video supplements

In preparation.

## 8 Author contribution

SEA coordinated efforts of various parties, carried out formal analysis, set up the salinity measurement field campaign and models, pre & post-processed and visualized the data and led the manuscript draft. PH and MVV were responsible for funding acquisition, supervision, and review of the research and manuscript. SEA, together with MVV and PH conceptualized the

study. HK and AVD supported software application and conducted the required numerical model development/update at various stages. HK contributed to the writing and reviewed the paper, and AVD reviewed the final draft. GVP carried out the field measurement bathymetry and prepared the bathymetry for application in the modelling exercise and reviewed and edited



the final manuscript. DDD provided resources and reviewed the findings and the writing, while NNT and TTQ supported the investigation, analysis, model set-up, and reviewed the findings and the writing. LB assisted during the field campaign, and

for pre-processing and post-processing of the data, led the video abstract development and reviewed the final manuscript. NNH facilitated and coordinated the salinity and bathymetry measurement field campaigns and reviewed the final manuscript. SED and DRP facilitated the funding acquisition for the bathymetry field campaign, supervised the bathymetry development and contributed to the writing of the final manuscript.

**Acknowledgments:** We would like to show are gratitude to Mr Le Quan amongst other colleagues at SIWRR for their humble

support during the field campaign and our stay in the city of Dai Ngai. Without their support, this study would not be possible. Special thanks to the personnel of the Southern Institute for Water Resources Planning (Ho Chi minh City, Vietnam), for openly supporting the project during its development. We are grateful to Dr Jasper RFW Leuven for his support through the toughest times of the field campaign. We sincerely appreciate how DHI supported our research by generously providing us with wind, water level and velocity data in the coastal waters.

**Conflicts of Interest:** The authors declare they have no conflict of interest.

**Funding:** This research is part of the "Rise and Fall" project, funded by NWO-WOTRO (W 07.69.105), Urbanizing Deltas of the World-1 (UDW1). The 2018 bathymetric data were acquired with support from NERC, under two distinct projects referenced NE/S002847/1 and NE/P014704/1.

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





**Figure 1: Digital elevation map of the Mekong Delta (Minderhoud et al., 2019) and the numerical domain developed for this study and the observation points and the studied cross-sections; a) an overview of the Mekong River Basin and the Mekong Delta (b) the salinity measurement trajectories along Dinh An and Tran De channels (red dotted lines) and the names of estuarine distributary channels (c) top-view of part of the 2DV-3D grid**



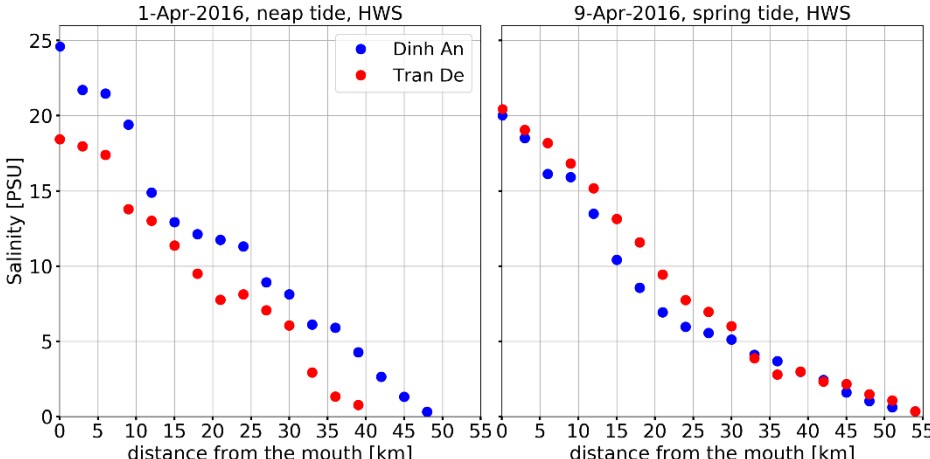

**Figure 2: Observed salt intrusion, during two consecutive neap (left) and spring (right) tides along the two lower Song Hau distributary channels (Dinh An and Tran De), during the 2016 VMD drought event**



Earth **Surface**
Dynamics
Discussions

**Figure 3: Water level at Can Tho (80 km from the estuary mouth) and offshore ocean wind speed and direction (a), subtidal discharge at Can Tho (b), and stationary salinity measurements in the lower distributary channels of the Hau River, Dinh An channel (c) and Tran De channel (d) prior and during the field campaign.**

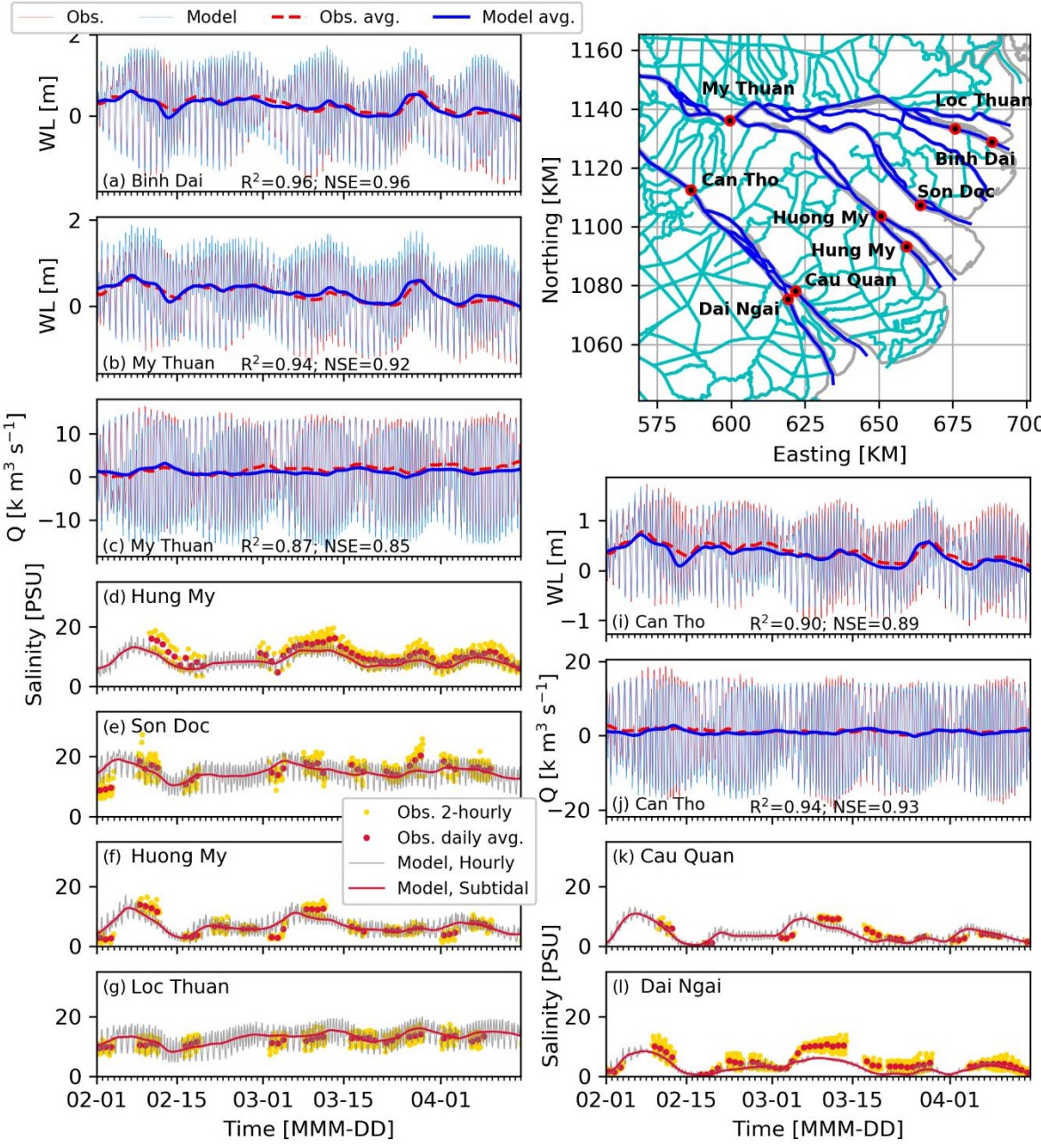

**Figure 4, Model results versus observed stationary water level, discharge and salinity measurements at multiple stations across the Mekong Delta**



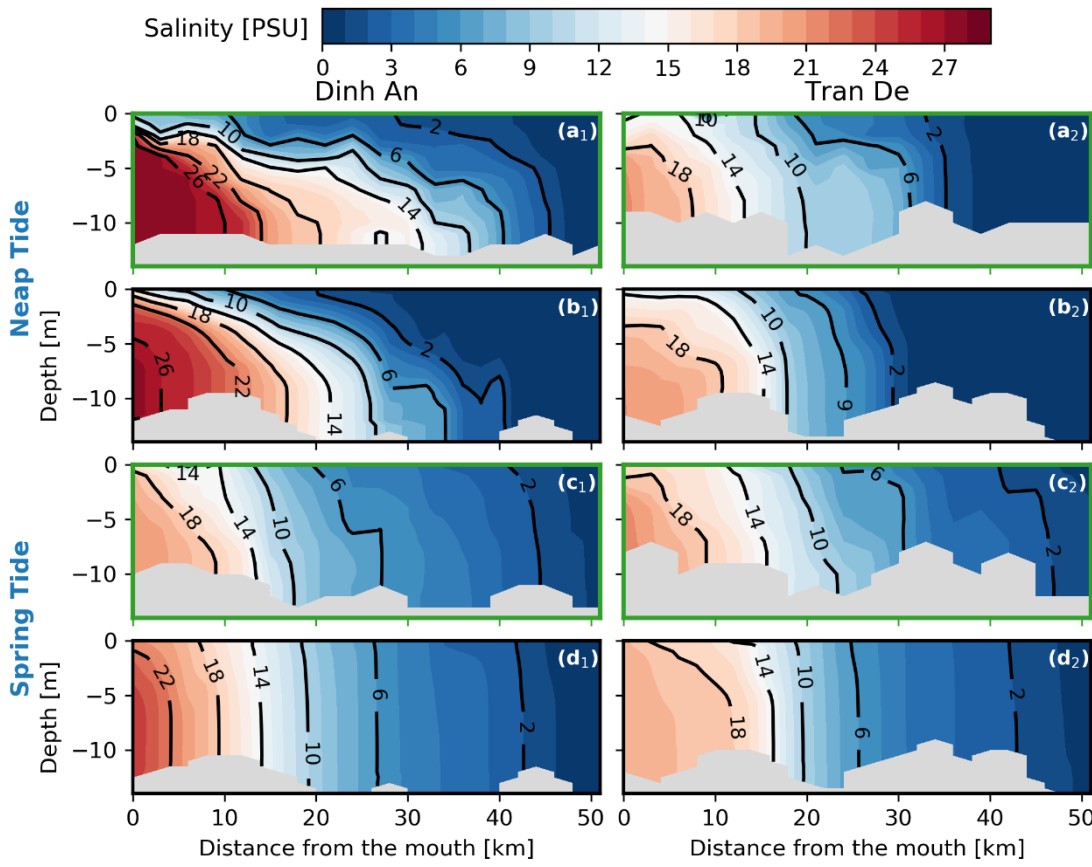

**Figure 5, Measured (a & c panels with green frames) versus modelled (b & d panels) HWS salinity structure along two distributary channels of the Hau River during Neap tide (a & b panels, Apr.-1st) versus spring tide (c & d panels, Apr.-9th) of the year 2016**



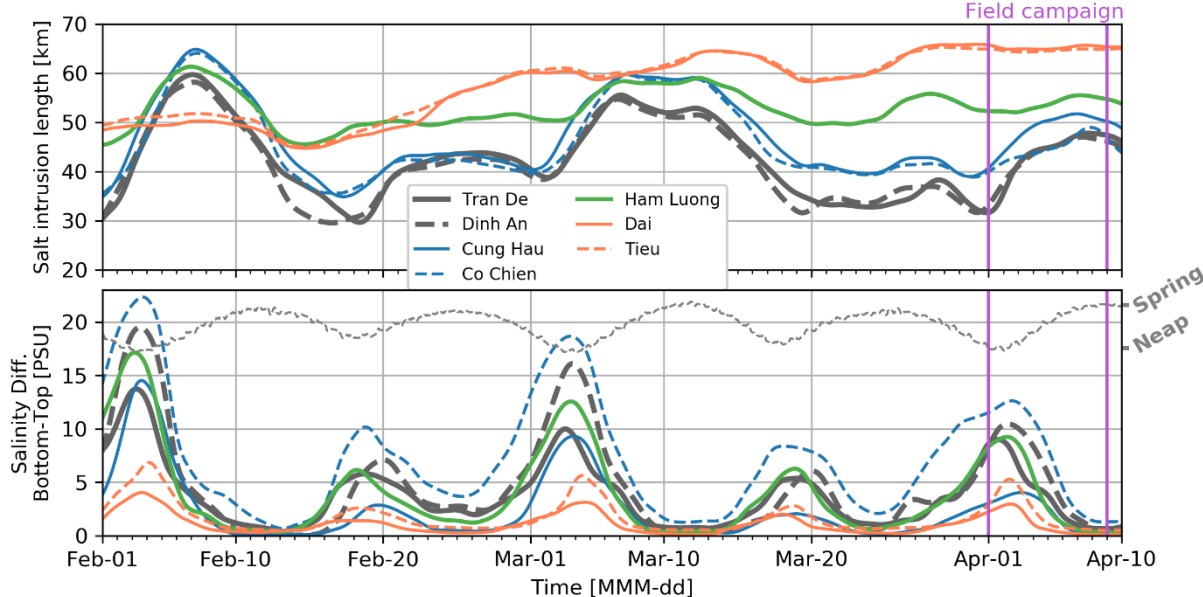

**Figure 6, Modelled salt intrusion length along seven estuarine distributary channels of the Mekong (upper) and over-depth salinity difference, at a point along the thalweg, ~15 km from each estuarine distributary channel (lower)**

Earth **Surface**
**Dynamics**
Discussions


**Figure 7, Comparison of the reference model (Ref.) including the ocean surge and the No-Surge (NS) model excluding the ocean surge for (a) water level (b) discharge; c & d show salinity in Dinh An and Tran De channels respectively; (e) salt intrusion length in time along the Dinh An distributary of the Hau River in the Reference (Ref.) model in comparison to NS (No-Surge), M2 (only including the M2 tidal constituent) and the M2S2 (replicating a spring-neap cycle) models; f & g show various salt fluxes (positive**

**upstream) in Dinh An and Tran De respectively at cross-sections 15 km from the estuary mouth in the reference simulation.**





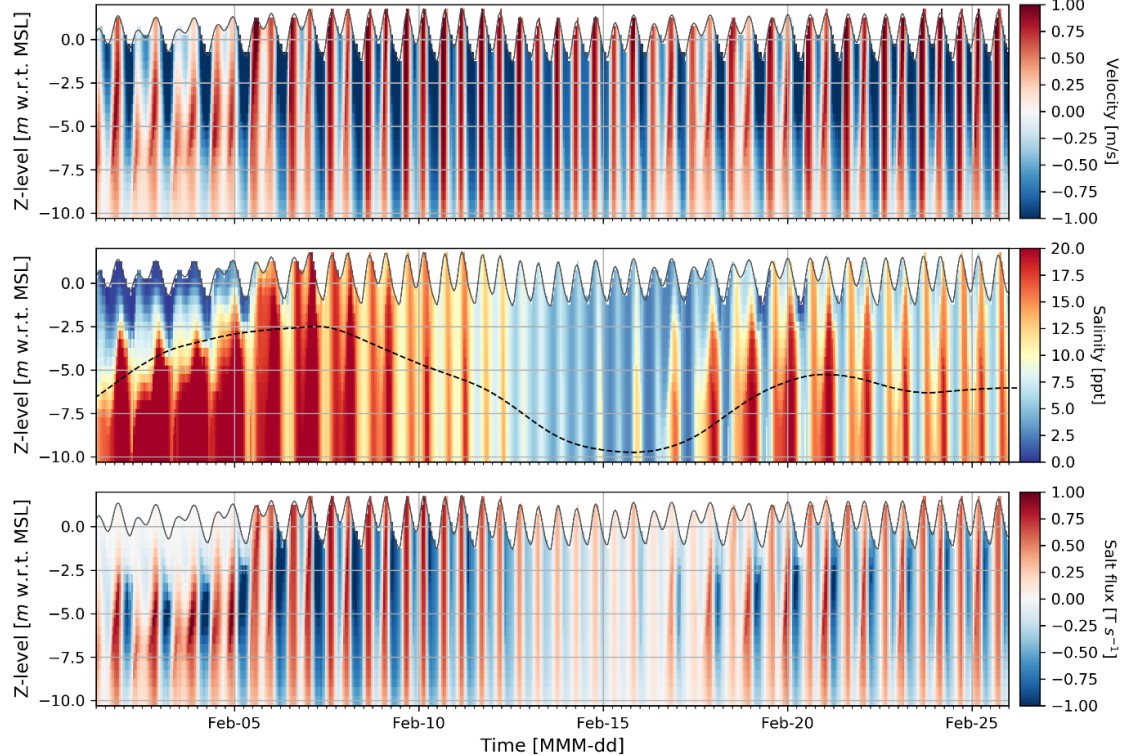

**Figure 8, Modelled temporal stationary estuarine variability at a point 15 km from the sea in the Dinh An channel in period-1 (maximum salinity); Upper panel: velocity profile in time; Middle panel: salinity profile in time (15 km from the sea) and subtidal depth-averaged salinity at Cau Quan in dashed black line scaled with the right y-axis (the right y-axis shares the labels with the colorbar); Lower panel: salt flux (product of salinity and velocity in a grid cell at a point along the thalweg)**


Earth **Surface**
**Dynamics**
Discussions

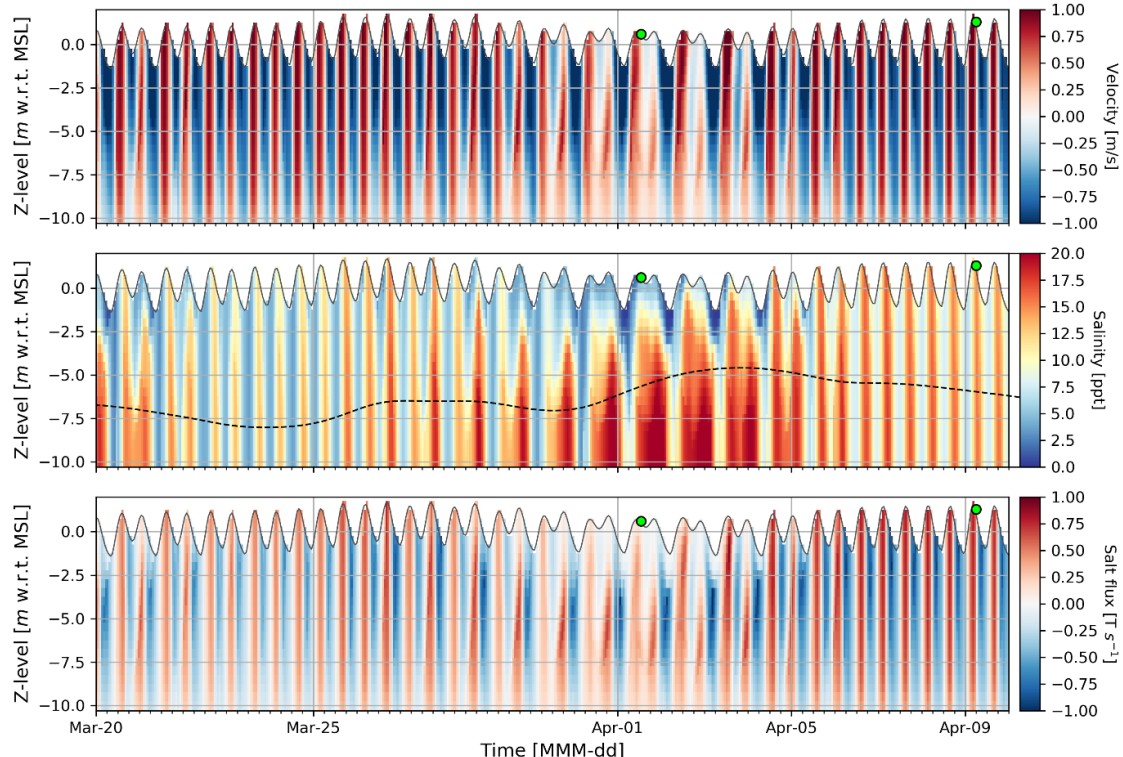

**Figure 9, Modelled temporal stationary estuarine variability at a point 15 km from the sea in the Dinh An channel in period-2 (field campaign); Upper panel: velocity profile in time with the green circles marking the measurement stances of the field campaign along the Dinh AN channel; Middle panel: salinity profile in time (15 km from the sea) and depth-averaged salinity at Cau Quan in dashed black line scaled with the right y-axis (the right y-axis shares the labels with the colorbar); Lower panel: salt flux (product of salinity and velocity in a grid cell, at a point along the thalweg)**




Earth **Surface**
**Dynamics**
Discussions

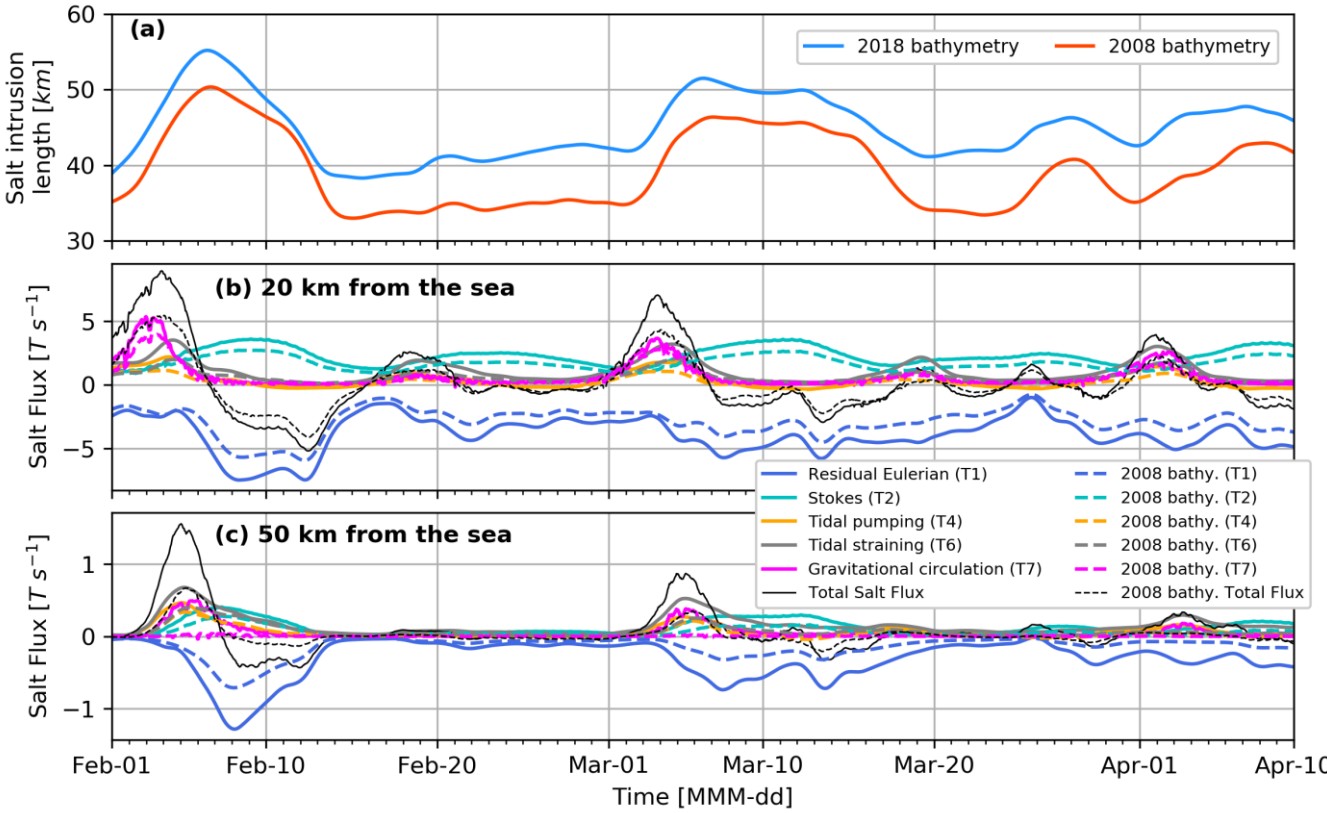

**Figure 10, Variation in tidally-averaged salt intrusion length in Ham Luong Channel under existing (2018) and the older (2008) bathymetries (a), Major salt flux mechanisms under two bathymetries at 20km (b) and 50 km (c) from the sea.**



Earth **Surface**
**Dynamics**
Discussions

EGU


**Figure 11,** Comparison of temporal variation of salt intrusion length along the Dinh An channel for models with (a) constant discharge including a spring-neap cycle (M₂S₂) and excluding spring-neap cycle with only semi-diurnal (M₂) or diurnal (K₁) tidal species; depicting the salt intrusion response along the Dinh An channel, in time and space to (b) a 10-day discharge perturbation from upstream (c) a 10-day subtidal offshore sea level perturbation when salt intrusion is in stationary conditions in models with
constant discharge excluding spring-neap cycles.

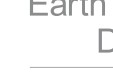
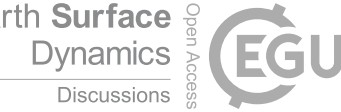

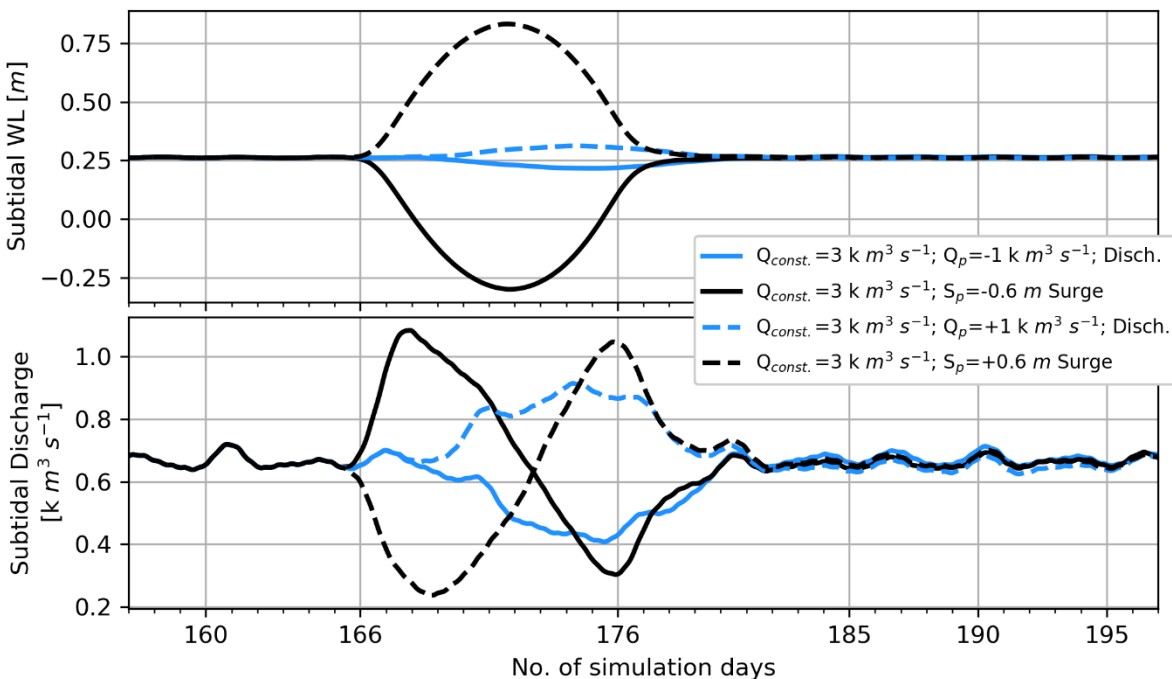

**Figure 12, Upstream (in Can Tho) subtidal water level and discharge response to variation in discharge and surge level**





**Table 1: Summary of input gauge, bathymetry, wind and tidal data**

| Input and validation data | Source | Description |
|---|---|---|
| 2008 bahymetry data | Eslami et al. (2019a) | data mainly from 2008 |
| 2016 bathymetry data | (Vasilopoulos et al., 2020) | The dataset obtained in 2018 covered the entire principle distributary channel network between the sea and Tan Chau/Chau Doc near the VN-Cambodia border (c180 km from the sea). |
| Water level data (My Thuan, Can Tho) | Collected by SRHMC and provided by SIWRP | Hourly time-series are with reference to Hondau (Vietnamese benchmark system) |
| Discharge data (My Thuan, Can Tho) | Collected by SRHMC and provided by SIWRP | Hourly time-series are from rating curves, generally updated once per 3-4 months, based on multiple days of continuous transect measurements |
| Stationary salinity observations | Collected by SRHMC and provided by SIWRP | Measured manually at different depths at an approximately fixed point in the estuary |
| Discharge in Kratie | MRC | Daily averaged discharge from rating curves |
| Offshore wind (40 km) | Climate Forecast System Reanalysis (CSFR), originally published by NCEP NOAA | The hourly wind parameters at 10 m, downloaded from online DHI metocean data portal |
| Evaporation | Literature | Considering reported values in the literature and in consultation with SIWRP |
| Water demand | SIWRP | Temporal variation as per (Eslami et al. 2019a) |
| Tidal constituents at the offshore boundary | TOPEX/Poseidon global inverse tide model (TPXO 8.0) | Thirteen leading harmonic constituents $M_2$, $S_2$, $N_2$, $K_2$, $K_1$, $O_1$, $P_1$, $Q_1$, MF, MM, $M_4$, $MS_4$, $MN_4$ |

**Table 2: list of simulations run in this study with a brief description**

| Models | Description |
|---|---|
| Ref. | Reference model with all the drivers (as described in section 2.4) |
| NS | (No Surge) excluding the offshore wind-driven subtidal water level |
| $M_2$ | All drivers, but offshore tidal constituent was limited to semi-diurnal tide, with $M_2 + 0.5$ $S_2$ amplitudes (no spring-neap variation) |
| $M_2S_2$ | All drivers, but offshore tidal constituent is limited to $M_2$ and $S_2$, developing a spring-neap variation |
| $K_1$ | All drivers, but offshore tidal constituent is limited to diurnal tide, with $K_1 + 0.5$ $O_1$ amplitudes; no spring-neap variation; minimum tidal mixing |



**Table 3, Summary of sensitivity analysis simulations and their descriptions**

| Models | Description |
|---|---|
| $M_2/K_1/M_2S_2$; $Q_{const.} = k\ m^3s^{-1}$ | 6 simulations characterized by their constant upstream discharge (3000 or 6000 m^3/s) and limited offshore tidal forcing. These are denoted by $M_2$ (single semi-diurnal tide, with amplitude $M_2+0.5S_2$), $K_1$ (single diurnal tide with amplitude $K_1+O_1$) and $M_2S_2$ (two semi-diurnal tides of $M_2$ and $S_2$, incl. spring-neap variability) |
| $Q_p = XX\ k\ m^3\ s^{-1}$ | 16 simulations with constant upstream discharge and subtidal water level, with a 10-day parabolic perturbation of XX $m^3\ s^{-1}$ magnitude from the constant discharge, 8 simulations with diurnal and 8 with semi-diurnal tide (excluding spring-neap variations) |
| $Q_s = XX\ m$ | 16 simulations with constant upstream discharge and subtidal water level, with a 10-day parabolic perturbation of XX m from the constant offshore subtidal water level, 8 simulations with diurnal and 8 with semi-diurnal tide (excluding spring-neap variations) |
