# Peer review of "Dynamics of salt intrusion in the Mekong Delta; results of field observations and integrated coastal-inland modelling"

_Earth Surface Dynamics, 2020_

## Referee Comment (RC1)

Dear Editor Prof. Passalacqua,

Thank you for sending me the manuscript "Dynamics of salt intrusion in the Mekong Delta; [...]", by Eslami et al. for review. I read the paper with great interest. The authors study the salt intrusion into the Mekong delta on hand of measurements and a numerical model. I like in particular that the authors show how bathymetric changes and surges influence the salinity intrusion, as well as how the dynamics in the smaller less connected distributaries differ from those in the more stratified main channels. The study appears by and large correct, but I have several comments worth a thought, and several minor comments that need clarification, and give some advice on how to improve the presentation. I recommend to accept the manuscript for publication, once it has been improved.

Kind regards,

**Comments**

- The simulations are insightful, and they could potentially contribute to a better general understanding of salinity intrusion, if the analysis, in particular, section 4.2, were given more theoretical depth.

  According to (*Savenije*, 2012), the salinity intrusion length $L$ can be estimated as:

  $$L = a \log \left( 1 + \frac{D_0 A_0}{K a Q_f} \right), \tag{1}$$

  Where $a$ is the convergence length, $D_0$ the dispersion coefficient, $A_0$ the cross section area at the inlet, $K$ van der Burgh's parameter and $Q_f$ the river discharge.

  - When do the assumptions fail on which the simple relation () is based on, i.e. how far and why is the 3D model really better and how could the simple relation be improved based on this insight?
  - How can the parameters do $D_0$ and $K$ be (better) estimated? In particular, how do they vary over the spring-neap cycle, during surges and change due to channel deepening through sand mining?

- The manuscript highlights the importance of on-shelf dynamics for the intrusion of salinity, yet the model only reaches about 70 km (c.f. line 144) into the sea, nowhere near the edge of the shelf which is at its closest point more than 250 km away. This has several limitations:

  - The model seems to account for the surge by raising the mean level at the sea boundary only, instead of modelling it directly. It would be interesting to model the surge fully through a larger domain on the shelf and barometric forcing. For the current setup, I am not fully convinced that a simpler 2DV model which only incorporates the river could not perform similarly well, as long as the surge level is applied at the river mouth in a similar fashion.

- The freshwater plume is very likely cut off and its dynamics not fully captured.
- Tidal waves will be spuriously reflected at the seaward boundary.

- Bedform dynamics vary the roughness on the seasonal and the spring-neap timescale and sand mining can reduce bedform dimensions and roughness. This might affect salinity intrusion as much as the deepening of the channels. The bed roughness employed in this study is rather crude. Similar ad-hoc approaches are typically taken in analytical and 1D model studies, but they are less justified for complex 3D models. As Delft3D can capture bedform migration and the induced roughness to a certain extent, it would be interesting to see this incorporated into the model.

- Suspended bed material transport introduces stratification and largely influences the turbulence structure. This certainly influences the mixing and hence the salinity intrusion. As sediment transport is a strongly non-linear process, the influence will be complex. As Delft3D-FM was primarily build to model sediment transport, it would be a pity not to apply it in the model.

- The study investigates the effect of bed level changes due to a reduction of sediment supply and sand mining. It is well known that this also causes wide-scale bank erosion and channel widening in the Mekong. How far does this influence the salinity intrusion and is this accounted for in the model?

- The study also investigates the effect of surges. Large surges potentially cause overbank flow. How far does this play a role in the MD and is this accounted for in the model?

- The study proclaims the superiority of 3D over 2D/1D models, yet the analysis and discussion are limited to 2D dynamics, namely the along channel and vertical dimensions. If the dynamics along the third dimension (across-channel) is relevant, then it should be analyzed and discussed, if not, 3D should be better be called 2D-vertical.

- It is redundant to provide both the $R^2$ and NSE as both have nearly identical definitions and limitations with respect to evaluating the goodness of fit. I suggest stating only the $R^2$. It would be much more interesting to provide the goodness of fit for the salinity, even if if it will be much poorer compared to the tides. This will give readers get an impression, what goodness of fit they could expect for their own studies.

**Minor**

"the continental shelf is an intrinsic component of the delta"
I disagree since the continental shelf is considerably larger than the Mekong (pro-)-delta, a more precise formulation would be "hydrodynamic processes on the continental shelf influence those in the delta".

"The delta's estuarine system is also more sensitive to upstream discharge variations."
Is this Mekong really more sensitive to river discharge variations than other rivers, why?

"in absence of [...] measurements [...], any prognostic evaluation of estuarine systems resorts to numerical modelling"
I disagree. In my experience complex 3D models do not perform well without extensive calibration and accurate bathymetric survey, and even when such data is available, simple empirical predictors fit to the observations often outperform complex models. However, I agree, that 3D models can provide insight into general dynamics and serve as a nifty way to interpolate and extrapolate observation in time and space.

Is there no better data than Milliman and Farnsworth? The authors seem to have access to an extensive dataset which should make them able to give more insight into the typical wet and dry season discharge. If not, it should be at least specified what the 350-500 indicate, i.e. interquartile ranges, etc.

slack tide : I think this means high water slack, please be more concrete.

"open source" : This is an overstatement. While the source code of the computational core of Delft3d-FM is indeed open source, the GUI which is necessary to set-up Delft3D-FM models is proprietary.

"Shallow Water Equations" : This is imprecise, Delft3D solves the (hydrostatic) Reynolds-Averaged Navier-Stokes Equations for the 3D parts of the model, and the SWE for the 2D and 1D parts.

Evaporation ... Is evaporation important for salinity in the Mekong Delta that it needs to be mentioned here? If so, does it mean evaporation from the water surface or the land surface?

bi-weekly : Alternating spring tides of different character are typical for deltas with mixed diurnal-semidiurnal tides. The observation, that this seems to strongly influence the salinity intrusion, deserves more attention. Does the difference in intrusion only stem from the difference in the tidal range or is the tidal asymmetry important which determines the shape of the wave?

Section 2.1 should be complemented with some basic information about the tides, at least state the mean spring tidal range or similar.

"During spring-tide, Stokes transport generates an upstream water level gradient that releases discharge towards the neap tide." :
Not quite. The oscillation of the subtidal water level offset over the spring-neap cycle is caused by higher friction during spring than during neap-tide. While this also causes the flow to oscillate over the spring-neap cycle, this is different from the Stokes transport, i.e. the transport which is caused by the phase difference between tidal velocity and surface elevation.

3.6 Salt fluxes : The description of how the salt fluxes are calculated is confuse and imprecise. Mathematical definitions are mixed with their discrete implementations, for the latter, grid, volumes and faces are mixed up. Note that also some form of interpolation is necessary, as fluxes are computed by Delft3d at volume faces, while salinity concentrations are computed at volume (ortho)-centres. I suggest to state only the mathematical definitions, and not to go into detail of the discrete calculation.

1.5 m incision : How deep was the channel, i.e. how large was the incision relative to the depth?

blocking river discharge : Imprecise, when a surge comes in, it superimposes a landward current, when it leaves, it superimposes a seaward current. The river discharge is not "blocked".

mixing parameters : Does this mean the diffusion coefficient? If so, it should be properly defined.

"The ocean surge ..., its impact can be twice as large as the surge period itself." : This seems to be rather coincidental because the simulated surge lasted for 10 days. It might better to write, the impact of the surge is felt even three weeks after the surge. Three weeks also seem quite long to me.

Data : Please provide at least a slimmed-down version of the model, i.e. the Delft3D-FM input files. Note that "can be provided" upon request for the data is not sufficient. Standard is providing data in a DOI-referenced repository or in the supplementary.

**Presentation**

Figure 1 Please mark Phnom Penh and the channel connecting to Tonle Sap on the map.

Figure 1 It is hard to make the grid out inset c. Consider to improve or remove it.

Figure 6 It would be insightful if the (weekly filtered) river discharge were plotted alongside the salinity intrusion, similar to the tidal range.

Figure 2 It would be insightful if this plot were accompanied by the modelled HWS and LWS salinity data for the same dates. This will allow for further insight regarding the classical theory of salt intrusion.

Figure 7 This plot is overloaded, g) has 9 lines. I suggest to provide a bar plot that provides average values for each for the fluxes for spring and neap. At least, try to limit the plot to just say the four most relevant fluxes, and combine the remaining fluxes into one residual flux.

Figure 10 This plot is overloaded, try to focus on what is relevant and the message you want to convey.

I would like to see a figure which shows the variability of the intrusion length - discharge relation, in particular in comparison to a simpler method, like the relation by Savenije .

**Suggested Textual Improvements (no reply expected)**

The text is understandable but needs a lot of minor corrections before being published, in many places, it misses essential articles or uses improper preposition. The text also mixes past and present tense, sometimes in a single sentence (c.f. 159 : Neuman [sic] conditions were ... wind is). In many places, the past tense gives the impression that the presented results were already outdated. I would personally use present tense wherever possible.

"upstream discharge variations" → "variations of the river discharge"

"is far more vulnerable" → "has become far more vulnerable"

VMD : Why not just MD?

makes → make

500 G m$^3$ → 500 10$^9$ m$^3$

seven : I count 8 when looking at the map, but this is probably a matter of definition tidal difference → tidal range?

in the delta → into the delta hr → h scheme → grid

Martyr-koller → Martyr-Koller in simulation year of 2016 → in the simulated year 2016

The Neuman conditions → Neumann conditions our field campaign failed to measure → we did not measure

At a sea → At the sea in all stations → at all stations is able to represent → represents can develop → reproduces

To study SWI → To study the SWI

experiences variable SWI → experiences a variable SWI

The lower [...] distributary channels follow similar trends of SWI → SWI follows similar trends in the lower distributary channels upstream discharge variations → variations of the river discharge.

channel network → channel (When only the Hau channel is further analyzed, then it is not any more a network.)

of surge on substantial → of the surge on the substantial

Upstream of Dinh An [...] network → Upstream of the Dinh An [...] network in subtidal discharge → in the subtidal discharge subtidal stationary salinity → tidally averaged salinity?

increase subtidal salt intrusion → increase the subtidal salt intrusion of 11 other → of the 11 other, or better → in absence of other tidal constituents, since there are infinitely many more show estuarine → show the estuarine

Period-1 → First period and second period sound more natural higher discharge → higher river discharge

[partially] → partially starts declining → declines?

average salinity → tidally averaged salinity?

flux in the downstream direction → flux into the downstream direction increasing total → increasing the total over-depth → depth-averaged elevation → surface elevation

As the flux calculation is carried out on numerical model results → As the flux is calculated from the numerical model output

As velocity → as the velocity constant within a numerical grid → constant on the surfaces of the finite volumes grid cell → finite volume discharge through a grid cell → discharge through a volume surface (Note that for this, the water surface elevation needs to be interpolated from the neighbouring volume centres to the face.)

decomposed to eight → decomposed into eight across the cross-section → across the river nearly double in width → nearly twice the width larger by nearly two fold → nearly twice as large have influenced → influence to other → to the other

That study → This study that tidal amplitude → that the tidal amplitude this value increases to 3 m → incision is up to 3 m

Stokes transport is maximum → Stokes transport is at its maximum is minimized → is minimal that is often → which is often

We showed → we show have dominant role in SWI into the VMD → dominate the SWI into the MD

process → processes ocean surge → ocean surges influence discharge division → influence the discharge division

While discharge variation inversely changes salt intrusion → While river discharge reduces salt intrusion intrusion , → intrusion, how exactly ocean surge → how exactly an ocean surge increasing channel depth → increasing water depth in the delta → into the delta

"the continental shelf ... is an intrinsic component of the delta system" Rephrase, processes on the shelf influence the delta, but they are in a geological sense not part of it. The shelf is much larger than the delta.

significantly → considerably significantly → considerably has provided → provides could reproduce → reproduces effect on discharge → effect on the river discharge twice longer → twice as long as

Figure 5 a & c panels → a and c, b & d panels → b and d

Figure 5 Apr.-1st → 1 April 2016, Apr.-9th → 9 April 2016

Figure 6 upper → a, lower → b

Figure 8  velocity profile → vertical profile of the streamwise velocity

Figure 8  in time → over time

Figure 8  product of salinity and velocity in a grid cell at a point along the thalweg → product of salinity and flow velocity at the thalweg

Table 3  $Q_s = XXm \rightarrow Q_s = XXm^3s^{-1}$ ?

Table 3  k → $10^3$

**References**

Savenije, H. H. G., *Salinity and Tides in Alluvial Estuaries, 2nd completely revised edition*, salinityandtides.com, 2012.

---

## Author Comment (AC1)

Dear editor,

We thank you for your time and for the opportunity to respond to the two detailed and constructive reviews. We specifically appreciate the raised discussion points that are for sure beneficial to a larger audience that can include hydrologists, estuarine scientists and physical geographers/oceanographers who study the larger deltaic systems dynamic. Based on the comments we have changed the manuscript when needed, and in a few cases, we decided to not make changes, but for that, we provide our reasoning.

We thank both reviewers for their invaluable time and contribution. From this point on, we respond point by point to the comments/suggestions in two sections separated per reviewer's comment letters.

**Yours Sincerely,**
**Sepehr Eslami, on behalf of the co-author's**

**Reply to comments by referee #1**

The simulations are insightful, and they could potentially contribute to a better general understanding of salinity intrusion, if the analysis, in particular, section 4.2, were given more theoretical depth. According to (Savenije, 2012), the salinity intrusion length L can be

estimated as:

$$L = a \log \left( 1 + \frac{D_0 A_0}{K a Q_f} \right),$$

Where a is the convergence length, $D_0$ the dispersion coefficient, $A_0$ the cross section area at the inlet, K van der Burgh's parameter and $Q_f$ the river discharge.

- When do the assumptions fail on which the simple relation () is based on, i.e. how far and why is the 3D model really better and how could the simple relation be improved based on this insight?
- How can the parameters do $D_0$ and K be (better) estimated? In particular, how do they vary over the spring-neap cycle, during surges and change due to channel deepening through sand mining?

Reply: It is indeed insightful to compare the 3D model results with existing relations, because very often due to data availability, there are limitations to 3D/sophisticated numerical applications. We have therefore made a thorough comparison to application of the Savenije 2005 formulation. We made the comparison for the Dinh An channel. First we fit the analytical model to the field measurements during spring and neap tides, then made a comparison between subtidal (tidally-averaged) numerically versus analytically modelled salt intrusion length and stationary observations at a point 27 km from the sea.

The results showed that the analytical model can reproduce the intrusion length and its profile relatively accurately (as has been shown before). As the analytical model overestimates intrusion length during neap tide, the numerical model underestimates intrusion in the same period. Both models do reasonably well at spring tide. Note that the estuary at the mouth is a critical input in the analytical model to develop the right profile, while the 3D model predicts the river-ocean interaction at the mouth relatively accurately. In terms of temporal variation, the analytical model has a direct response to upstream discharge, therefore it basically follows the observed subtidal discharge variation, which at times, when upstream subtidal discharge changes [momentarily] significantly intrusion length can also be largely mis-calculated. Therefore, while it still produces the orders of magnitudes reasonably well, in reality, the estuary is always in adaptation and some of these adaptation processes cannot be captured by analytical models in a temporal manner. However, we maintain that when it comes to general trends in the estuarine systems, the analytical models can be extremely valuable as they can accurately inform on the system response to specific drivers. But when it comes to temporal variations, application of a numerical model may be inevitable. As we had to limit the scope of this study, we strongly encourage other scientists to carry out a similar comparison between the 3D model results and a 1D model of the delta and report the range of applicability of the 1D models, given their limitations (e.g., at reproducing salinity at the estuary mouth).

The manuscript highlights the importance of on-shelf dynamics for the intrusion of salinity, yet the model only reaches about 70 km (c.f. line144) into the sea, nowhere near the edge of the shelf which is at its closest point more than 250 km away. This has several limitations:

- The model seems to account for the surge by raising the mean level at the sea boundary only, instead of modelling it directly. It would be interesting to model the surge fully through a larger domain on the shelf and barometric forcing.
- For the current setup, I am not fully convinced that a simpler 2DV model which only incorporates the river could not perform similarly well, as long as the surge level is applied at the river mouth in a similar fashion.
- The freshwater plume is very likely cut off and its dynamics not fully captured.
- Tidal waves will be spuriously reflected at the seaward boundary.

Reply: The reviewer raises some interesting points, but they are not of a concern in this case. The shelf is large enough to capture the main dynamics. We do indeed 'prescribe' the surge and not solve it, but this is sufficient. By describing it as a subtidal wave at the offshore boundary, including a time difference to account for the propagation from the boundary to the mouth of the VMD, we capture the effects of the surge without having to model surge dynamics in the entire South China Sea. Note that to re-develop the surge signal, the model domain needs to be far beyond the continental shelf and as large as the South China Sea. The match between observed and modelled subtidal water levels is quite good all over the delta and as far as 250 km inland, so this approach is sufficient.

A 2DV model might perform equally well, but there would be issues at the mouth because one would have to prescribe the salinity. The freshwater plume at sea is also influencing the salinity dynamics in the estuarine channels. Maybe it would be sufficient to model the estuaries in 2DV connected to the 3D shelf sea. We did make some first efforts in such an approach but experienced challenges regarding ocean-river interaction at the mouth and abandoned those efforts at some point. While we highly appreciate the idea of a model as simple as possible, in this case it was not sufficient, and we encourage further effort in advancing the 2DV numerical solution for estuarine channels as wide as Mekong (2-3 km).

The freshwater plume is very much contained within the model and generally travels south along the coast. This is mainly due to the southward monsoon driven flow (interacting with the tidal signal) that brings freshwater and sediment towards the Ca Mau Peninsula and out of the model domain. Note that this means that the effect of freshwater plume of the Northern channels (e.g., Ham Luong,) are then present for the more southern channels (e.g., Co Chien, Dinh An, etc.).

We used a combination of prescribed water levels and Neuman conditions (water level gradient). While there could still be spuriously reflected tidal waves, the good match between modelled and observed tidal and subtidal signal eliminates concerns on model behaviour.

Bedform dynamics vary the roughness on the seasonal and the spring- neap timescale and sand mining can reduce bedform dimensions and roughness. This might affect salinity intrusion as much as the deepening of the channels. The bed roughness employed in this study is rather crude. Similar ad-hoc approaches are typically taken in analytical and 1D model studies, but they are less justified for complex 3D models. As Delft3D can capture bedform migration and the induced roughness to a certain extent, it would be interesting to see this incorporated into the model.

Reply: The bed material in the Mekong is mainly fine material and some sand (Gugliotta et al., 2017). We agree that bed forms influence the roughness and thereby the vertical mixing. However, adding these mechanisms is really beyond the scope of the present paper. First, it is highly uncertain how bedforms change as a function of hydrodynamic forcing. In Delft3D the van Rijn predictor is included, but it is not necessarily always functioning very well (Brakenhoff et al., 2020). For example, it overpredicts the variability in time of bed forms.

Secondly, there are no bedform predictors that can handle the mixed sediments. In complete sandy environment the bed forms can develop, but at a certain critical transition the bedforms cannot form anymore. This is still highly uncertain. Therefore, we think that using a bedform predictor would not improve the predicted salinity. It would mainly add complexity, computation time and probably the prediction would not improve. In fact, most 3D models still use roughness as a calibration parameter. We followed the same approach and calibrated manning's $n$ value such that mismatch between observed and modelled water levels and discharge was minimized, therefore, we see the adopted approach as sufficient.

Suspended bed material transport introduces stratification and largely influences the turbulence structure. This certainly influences the mixing and hence the salinity intrusion. As sediment transport is a strongly non-linear process, the influence will be complex. As Delft3D-FM was primarily built to model sediment transport, it would be a pity not to apply it in the model.

Reply: This is again an interesting suggestion, but not so easy to implement. First, such a sediment transport model often needs significant calibration and validation exercise, and we are aware of such parallel efforts. However, we see the effect on general salinity dynamics as limited, while we do not rule out the possibility of its effect (if implemented accurately) improving the model performance. However, the question remains: When would it become important? From (McLachlan et al., 2017) we learn that during dry season top to bottom sediment concentration differences are up to 400 mg/l, which means $\frac{\Delta\rho}{\rho} = 0.0004$. Typical top to bottom salinity differences can reach 5 – 10 PSU. This induced typical density differences of delta $\frac{\Delta\rho}{\rho} = 0.008$, so at least a factor 10 larger. In this system, density differences are fully captured by salinity levels. During well mixed conditions it might become important, but that is typically not the case here.

The study investigates the effect of bed level changes due to a reduction of sediment supply and sand mining. It is well known that this also causes wide-scale bank erosion and channel widening in the Mekong. How far does this influence the salinity intrusion and is this accounted for in the model?

Reply: That is indeed a common consequence of sediment starvation within the delta. However, its effect on salt intrusion, at least in the short term and on small scale dynamics, is expected to be very limited. The estuarine channels are typically 1.5-4 km wide, while the order of magnitudes of riverbank erosions are in the order of a few tens of meters where they take place (factor 50-150 smaller). Therefore, also as the vertical processes are dominant, we expect limited effect on salinity dynamics at this stage.

The study also investigates the effect of surges. Large surges potentially cause overbank flow. How far does this play a role in the MD and is this accounted for in the model?

Reply: This indeed plays a role in the wet season. However, we entirely focused on the dry season when the discharge is so small that flooding is nearly negligible. Occasionally, tidal flooding does take place in some of the coastal provinces (e.g., in Ben Tre Province at the farms or village level), but its effect on tidal and salinity dynamics is too small to be relevant.

The study proclaims the superiority of 3D over 2D/1D models, yet the analysis and discussion are limited to 2D dynamics, namely the along channel and vertical dimensions. If the dynamics along the third dimension (across-channel) is relevant, then it should be analysed and discussed, if not, 3D should be better be called 2D-vertical.

Reply: We used a 3D model and not a 2DV model. The Mekong bathymetry is not having

large depth variations over the width (mostly a single channel and no large intertidal areas). Therefore, it is correct to conclude that the over-depth processes are the main drivers of salinity variation in time. However, the 3D processes become relevant at the estuarine mouth and junctions when multi-directional flow do take place with significant repercussions for salinity in the rest of the system. Otherwise, the tidal flow direction is mainly along the estuarine channels, except at times of the turning tide. We included the point in the discussion.

It is redundant to provide both the $R^2$ and NSE as both have nearly identical definitions and limitations with respect to evaluating the goodness of fit. I suggest stating only the $R^2$. It would be much more interesting to provide the goodness of fit for the salinity, even if it will be much poorer compared to the tides. This will give readers get an impression, what goodness of fit they could expect for their own.

Reply: While it is correct that $R^2$ and NSE provide partly similar information, but still, they provide complementary input. $R^2$ was used to represent the linear relation between observations and the model and NSE provides indication on goodness of fit. For example, two signals can have high $R^2$ with a large bias, but that kind of information can be shown in NSE. Regarding statistics of salinity, while we do find it educating to see what typical accuracy can be expected, we do not find the comparisons one-to-one and just. Apart from irregularity (gaps) of salinity measurements, other limitations, and complications (as stated in the methods section, such as the exact location and methodology) makes the comparison difficult and attaching a statistical parameter is more mis-leading than informing; or it may lead to misjudgment. We believe there is no harm in making a qualitative comparison when the parameter is as sensitive and dynamic as salinity in an estuarine system.

22 "the continental shelf is an intrinsic component of the delta"

I disagree since the continental shelf is considerably larger than the Mekong (pro—delta), a more precise formulation would be "hydrodynamic processes on the continental shelf influence those in the delta".

Reply: Agreed. We changed the text accordingly.

24 "The delta's estuarine system is also more sensitive to upstream discharge variations."

Is this Mekong really more sensitive to river discharge variations than other rivers, why?

Reply: It is more sensitive than some other systems as mentioned in the discussion section 4.2. We meant that as intrusion length is related to a power law of river discharge $L = \alpha \, Q_f^n$, with n=-0.5 (in Mekong), in comparison to other systems with n=-0.2 to -0.3, such as Hudson River, Skagit and Merrimack, the Mekong Delta has a quicker response time and larger response in terms of intrusion variations. This is perhaps driven by various drivers such as the along-channel large intrusion length in the dry season, energetic tidal dynamics (large tidal range) and low freshwater flow in the dry season.

34 "in absence of [...] measurements [...], any prognostic evaluation of estuarine systems resorts to numerical modelling"

I disagree. In my experience complex 3D models do not perform well without extensive calibration and accurate bathymetric survey, and even when such data is available, simple empirical predictors fit to the observations often outperform complex models. However, I agree, that 3D models can provide insight into general dynamics and serve as a nifty way to interpolate and extrapolate observation in time and space.

Reply: We agree that the statement may be too strong. We changed it into "prognostic evaluation of estuarine dynamics can significantly benefit from numerical modelling"

99 Is there no better data than Milliman and Farnsworth? The authors seem to have access to an extensive dataset which should make them able to give more insight into the typical wet and dry season discharge. If not, it should be at least specified what the 350-500 indicate, i.e. interquartile ranges, etc.

Reply: We added information based on the available data in section 2.1 (yearly total interquartile 340-450 $10^9$ m$^3$ and dry season nearly 7-9% of the total yearly discharge)

125 slack tide: I think this means high water slack, please be more concrete.

Reply: Corrected.

130 "open source" : This is an overstatement. While the source code of the computational core of Delft3d-FM is indeed open source, the GUI which is necessary to set-up Delft3D-FM models is proprietary.

Reply: One may set-up Delft3D-FM model without the GUI. However, currently, users can also receive a GUI under a beta-user agreement. So we consider this statement to be correct.

131 "Shallow Water Equations": This is imprecise, Delft3D solves the (hydrostatic) Reynolds-Averaged Navier-Stokes Equations for the 3D parts of the model, and the SWE for the 2D and 1D parts.

Reply: We changed this according to the reviewer's remark.

160 Evaporation ... Is evaporation important for salinity in the Mekong Delta that it needs to be mentioned here? If so, does it mean evaporation from the water surface or the land surface?

Reply: Yes, it is important, especially when it comes to freshwater budget and given the sensitivity of the delta/model to discharge. Without taking this into account the model overestimates available freshwater with some 4-6% (also partly addressed in Eslami et al., 2019) which translates to a larger error in salinity. Evaporation takes place over the wet surface of the model which includes all the main, primary and secondary channels as well as the ocean surface.

228 bi-weekly: Alternating spring tides of different character are typical for deltas with mixed diurnal-semidiurnal tides. The observation, that this seems to strongly influence the salinity intrusion, deserves more attention. Does the difference in intrusion only stem from the difference in the tidal range or is the tidal asymmetry important which determines the shape of the wave?

Reply: The tidal waves are not very asymmetric yet in the downstream parts of the estuary (M4 amplitude at Can Tho is about 15 cm). The asymmetry becomes more important upstream, but that part is completely fresh and tidal asymmetry does not influence the vertical mixing. The differences in salt intrusion length mainly seem to correlate with the tidal range, together with effect of subtidal water level and variations in river discharge.

97 Section 2.1 should be complemented with some basic information about the tides, at least state the mean spring tidal range or similar.

Reply: Information is added to the section 2.1.

250 "During spring-tide, Stokes transport generates an upstream water level gradient that releases discharge towards the neap tide.":

Not quite. The oscillation of the subtidal water level offset over the spring-neap cycle is caused by higher friction during spring than during neap-tide. While this also causes the flow to oscillate over the spring-neap cycle, this is different from the Stokes transport, i.e. the transport which is caused by the phase difference between tidal velocity and surface elevation.

Reply: We agree. We changed it into "During spring-tides the river-tide interaction results in larger subtidal friction, which causes the generation of a subtidal wave and temporary storage of water, that is released during neap tide. "

278 3.6 Salt fluxes: The description of how the salt fluxes are calculated is confuse and imprecise. Mathematical definitions are mixed with their discrete implementations, for the latter, grid, volumes and faces are mixed up. Note that also some form of interpolation is necessary, as fluxes are computed by Delft3d at volume faces, while salinity concentrations are computed at volume (ortho)-centres. I suggest to state only the mathematical definitions, and not to go into detail of the discrete calculation.

Reply: the text is simplified and limited to the mathematical definition.

324 1.5 m incision: How deep was the channel, i.e. how large was the incision relative to the depth?

Reply: We added 'The bathymetry was deepened with 1.5, which is typically about a 10% change in still water depth, depending on the location". This info is added to the text.

435 blocking river discharge: Imprecise, when a surge comes in, it superimposes a landward current, when it leaves, it superimposes a seaward current. The river discharge is not "blocked".

Reply: It is not only the current that matters, but also the water levels. The landward current also translates into a water level gradient and changes in water levels, so we still think we can use the word "blocking". It is a matter of taste. However, we modified the text into: "However, a rising surge causes an upstream directed flow, decreasing local discharge, while a falling surge drives a seaward directed flow, resulting in a rising discharge"

446 mixing parameters: Does this mean the diffusion coefficient? If so, it should be properly defined.

Reply: No, this relates to vertical mixing. They call it the 'mixing parameter', so we use their terminology. To make this clearer we added "… mixing parameter (their parameter M)".

474 "The ocean surge …, its impact can be twice as large as the surge period itself." : This seems to be rather coincidental because the simulated surge lasted for 10 days. It might better to write, the impact of the surge is felt even three weeks after the surge. Three weeks also seem quite long to me.

Reply: text changed to "The ocean surge, however, leads to a near-immediate response in SWI and because of its rebound effect on discharge, its impact was felt twice longer than the surge period itself". The surge impact is not felt three weeks after, but it's perturbation impact is nearly twice as long as the surge effect. We are not clear what is rather coincidental.

490 Data: Please provide at least a slimmed-down version of the model, i.e. the Delft3D-FM input files. Note that "can be provided" upon request for the data is not sufficient. Standard is providing data in a DOI-referenced repository or in the supplementary.

If the data can be provided to public, it can indeed be stored in a repository. As mentioned, parts of the model input are not open to public as they are considered the institutional property of the local Vietnamese institutes and we do not have their consent to publish those data. However, if the reviewer requests access to the model input, all the models and codes can be temporarily placed in a cloud for review. Furthermore, it is not clear what a slimmed down version of the model would mean. If this refers to model resolution, it cannot be simply reduced as we have different dimensions within the same model domain. This requires a separate effort to make an acceptable lower resolution model, which then cannot be used for salinity dynamic assessment. We have been discussing with our partners to accept an opensource approach for data, and we hope in the near future we are able to achieve that. Nevertheless, if parties are interested, it is possible to develop arrangements through which other institutes receive access to those data.

**Presentation**

Figure 1 Please mark Phnom Penh and the channel connecting to Tonle Sap on the map.

Reply: Done

Figure 1 It is hard to make the grid out inset c. Consider to improve or remove it.

Reply: We understand it may cause some complication in presentation, but still it is better to include the grid and its extent than having it removed. In Deflt3D, it used to be possible to reduce the grid resolution for presentation, but as the grid administration in Delft3D-FM is different, it is not yet possible to do that. If removed, it takes whole lot of text to explain and visualization is going to be difficult. Furthermore, we have used this figure for various presentations, and we normally have positive feedbacks in terms of readability. So, we prefer to keep the grid for educational purposes.

Figure 6 It would be insightful if the (weekly filtered) river discharge were plotted alongside the salinity intrusion, similar to the tidal range.

Reply: Subtidal discharge is scaled and added to the figure.

Figure 2 It would be insightful if this plot were accompanied by the modelled HWS and LWS salinity data for the same dates. This will allow for further insight regarding the classical theory of salt intrusion.

Reply: the modelled values are now added to Figure 13 where we compare the analytical and numerical models.

Figure 7 This plot is overloaded, g) has 9 lines. I suggest to provide a bar plot that provides average values for each for the fluxes for spring and neap. At least, try to limit the plot to just say the four most relevant fluxes, and combine the remaining fluxes into one residual flux.

Reply: We prefer to show the time behavior, because it suggests that the balances change in time and that is key to explaining the estuarine variability. We like the idea of only showing the dominant fluxes, but the problem is that only T3 and T5 (two of the tidal pumping fluxes) and T8 (triple correlation) are negligible. Other than these, T4 (tidal pumping) is the smallest that often in the neap tide reaches up to 30% of the total flux. We, therefore,

removed T3, T5 and T8 from the graph and left a remark in the legend.

Figure 10 This plot is overloaded, try to focus on what is relevant and the message you want to convey.

Reply: Although there are many lines, the Figure nicely shows how the change in bathymetry affects the different salt transport contributors. We do not see how we can simplify this plot. The focus of the figure is showing the changes in fluxes in response to riverbed level changes.

I would like to see a figure which shows the variability of the intrusion length - discharge relation, in particular in comparison to a simpler method, like the relation by Savenije.

Reply: See our earlier reply and Figure 12.

**Suggested Textual Improvements (no reply expected)**

The text is understandable but needs a lot of minor corrections before being published, in many places, it misses essential articles or uses improper preposition. The text also mixes past and present tense, sometimes in a single sentence (c.f. 159: Neuman [sic] conditions were ... wind is). In many places, the past tense gives the impression that the presented results were already outdated. I would personally use present tense wherever possible.

Reply: We made a careful check of the new text and tried to be consistent.

24      "upstream discharge variations" −−− "variations of the river discharge" Changed

33      "is far more vulnerable" −−− " has become far more vulnerable" Changed

80      makes --- make Changed

99      500 G m3 --- 500 109 m3 Changed

101     seven : I count 8 when looking at the map, but this is probably a matter of definition

7 channels are identified as estuarine as named in Figure 1b, the 8th one is currently closed to the sea.

113     tidal difference --- tidal range? Changed

114     in the delta --- into the delta Changed

125     hr --- h Not changed

135     scheme --- Not changed. Grid scheme as in a methodology (opposing to sigma layer scheme)

140     Martyr-koller --- Martyr-Koller Changed

156     in simulation year of 2016 --- in the simulated year 2016 Changed

159     The Neuman conditions --- Neumann conditions Changed

175     our field campaign failed to measure --- we did not measure Changed

194    At a sea --- At the sea Changed

210    in all stations --- at all stations Changed

218    is able to represent --- represents Changed

219    can develop --- reproduces Changed

222    To study SWI --- To study the SWI Changed

224    experiences variable SWI --- experiences a variable SWI Changed

225    The lower [...] distributary channels follow similar trends of SWI

--- SWI follows similar trends in the lower distributary channels Changed

236    upstream discharge variations --- variations of the river discharge. Changed

244    channel network --- channel (When only the Hau channel is further analyzed, then it is not any more a network.) Changed

248    of surge on substantial --- of the surge on the substantial Changed

248    Upstream of Dinh An [...] network --- Upstream of the Dinh An [...] network Changed

250    in subtidal discharge --- in the subtidal discharge Changed

254    subtidal stationary salinity --- tidally averaged salinity? Subtidal, or tidally averaged salinity is calculated similar to discharge and WL subtidal signals, so we don't see the reason to change it (for sake of consistency).

254    increase subtidal salt intrusion --- increase the subtidal salt intrusion Changed

**Reply to comments by anonymous referee #2**

The Field Campaign section could be expanded a bit more. In particular, the depth of salinity measurements and the conversion method from conductivity to PSU should be discussed. To improve readability, along with the citation, the moving boat measurement technique should be briefly highlighted in this section.

Reply: we have made a few changes in the methods section and explained the method in a clearer manner. See section 3.2.

The phrases "saline water intrusion," "salt intrusion," and "SWI" are used seemingly interchangeably throughout the manuscript. Please use "saline water intrusion (SWI)" (or something similar) at the first occurrence of the phrase then only use SWI throughout the remainder of the paper to improve clarity. If saline water intrusion and salt intrusion are intended to highlight different processes highlight the difference.

Reply: We have unified the term SWI throughout the manuscript.

To many people, saline water intrusion or saltwater intrusion refers to high salinity in groundwater aquifers as opposed to surface water (for example, Todd, 74; Bobba 2002; Michael et al 2013). Please point out early in the manuscript that the focus of this study is surface water processes. With that being said, information on groundwater salinity in the shallow aquifers of the delta could provide some valuable context to readers on potential groundwater endmembers. If such data is available, I recommend at least a mention of the typical ranges of salinity in groundwater relative to surface water.

Reply: We made this now immediately clear in the introduction and added a reference to paper on salinity in ground water system.

The authors suggest that 3D dimensional models perform better for the Mekong Delta compared to 1D and 2D models. This may be the case, but without any direct comparison of this impressive 3D model with simpler 1D or 2D models, readers are left to assume that this is indeed the case. I suggest comparing these results with at least 2D models if they are available.

Reply: Also based on the comments made by reviewer 1, we decided to compare with the predictions based on Savenije (2005) analytical model (see section 4.3). Indeed, we may have not been very clear on where we see the advantages of the 3D model. We have modified the manuscript to more clearly define where we see the differences (e.g., see lines 65-70 introduction). When referring to 2D models, while there is a case to be made for application of 1D and 2DV models, we believe 2DH models simply cannot be applied for saline water intrusion modelling. There has been some applications of that (referred to in the introduction and discussion e.g., Tran Anh et al., 2018), but there is no physical explanation for application of a 2DH models. However, the authors started to apply a 2DV model, but found it difficult to have an accurate interaction at the estuary mouth with the 3D domain of the continental shelf. 1D models, with some definition of dispersion coefficient (e.g., Thatcher Harlemann) can develop decent trends of temporal variation of salinity. However, if we solve the problem of subtidal processes (e.g., ocean surge) by measured timeseries, the challenge remains on salinity of the lower boundary (estuary mouth) that cannot be computed based on physical processes and remains a function of certain assumptions. Therefore, we believe, depending on the application, 1D models and 2DV models (if properly tested) can be decent alternatives, but the accurate image (e.g., for salinity forecast) demands a 3D approach.

Finally, are the authors modelling the whole continental shelf? If not, mention that only a portion of the shelf is being modelled.

Reply: Although a relatively large part, we are indeed modelling a portion of the shelf. We changed this in the revised manuscript.

**Specific Comments**

Line 33: Is the delta more vulnerable compared to other deltas or more vulnerable compared to past versions of the delta?

Reply: We meant compared to the past version. We added 'than previously' to this sentence.

Lines 39-42: "Eslami et al., (2019b) showed that there has been increasing trends of salt intrusion and tidal amplification… that are being driven by bed level changes…" There seems to be enough room in the introduction to breakdown the link between SWI and bed level changes. Where are the bed level changes occurring? Are they occurring at the mouth, upstream, within the delta channels? Briefly point out how those changes affect tidal amplitude and SWI.

Reply: In principle, these riverbed level changes more or less take place all over the estuarine network with some exceptions. While we cannot show very accurately where the differences are as comparison to the older bathymetries is not straightforward, we are aware of other colleagues working on that. However, our 2019 paper (Eslami et al., 2019b) estimated typical depth changes of on average 2-3 m. We also managed to reproduce tidal propagation speeds which clearly indicated that the incisions have taken place mainly half way the estuarine distributary channels and towards upstream tidal rivers (see the supplementary information of that paper). A recent submitted paper by Vasilopoulos et al.( 2020) discusses further details on bed level changes. We have also added an additional sentence and more citations on that in the introduction.

Line 68: Cite examples of some other deltas around the world. If relevant, perhaps cut the following line "There are a number of field measurements…" and add the citations to line 68-69

Reply: modified the text and added some more references.

Line 85: Please elaborate on what is meant by "address estuarine variability." Do the authors mean to characterize 3D variability in salinity within the VMD?

Reply: modified.

Line 116: Here or earlier are good places to point out that this paper exclusively looks at surface water. Modified here and in the first paragraph of the introduction.

Lines 121: Please clarify how measuring SWI in 2016 gives insight into historical events.

Reply: Because this was event was a historical event and the underlying dynamics could be related to past and future changes. We agree it is not clear as it is written now. We changed it into: "To measure the dynamics during this extreme condition, we aimed to measure … "

Line 122: I think the paper would benefit from a brief summary of the moving boat measurements to supplement the references.

Reply: We have elaborated on the method in section 2.2 but note that the method is very simple in terms of procedure. So apart from the modified steps (as added), preparation and the equipment, there is limited to add. On its theory and why this is an interesting method, (Savenije, 1989 & 2012) can be referred to.

Line 126 What depth were the samples taken at?

Reply: information added (over the entire depth at 1 m resolution)

Line 126: What equation/method was used to transform salinity?

Reply: (UNESCO, 1981) and the information is added to the section.

Line 181: What was the SWI length in the Tran De channel and how did salinity near the mouth compare over spring and neap tide? These measurements for the Dinh An channel were compared but are not reported here for the Tran De channel.

Reply: This can be deduced from Figure 2 and also Figure 5. But note that the focus on Dinh An channel is simply because this channel was previously measured by (Nguyen and Savenije, 2006). We added "(see Figure 2)" to the sentence.

Line 203: Why was the sample location not a fixed point? Was it possible to erect some sort of semi-permanent structure like a stake into the bed?

Reply: It is done manually and depends on where the person can go at that moment relative to the target coordinate. Although it is not a fixed point, it is typically close (say within 30-50m) considering the channel width of 2-3 km.

Line 204: Manual measurements of salinity? Using a probe, grab samples. Really briefly remind the readers here how you are collecting these samples, so they do not have to go looking through the methods again.

Reply: Manual measurement here refers to the measurement carried out by the local authorities or hydro-met offices that carry out these measurements and provide the data.

Line 268: Higher to the field campaign (insert the date(s) of the field campaign here) to help with readability

Reply: We changed the text.

**Technical Corrections**

Line 37: Perhaps not "the key to land use" but rather the key to productive land use.

Reply: Not necessarily. Retreat can also be considered as an adaptation strategy.

Line 100: Point out the first split or remove "again."

Reply: Changed it.

Line 121: Consider something like "To interpret/estimate historical events we measured maximum …." Text is modified.

Line 123: Keep consistent: Either Apr or April. Modified

Line 123: Consider restructuring the sentence so that it reads something like "We simultaneously measured salinity structures and intrusion lengths along two channels over spring and neap tide (April 1 and 9, 2016, respectively)."

The term simultaneous refers to timing of measurements by two boats simultaneous along the two channels.

Line 125: Consider: "We sailed upstream from the estuary mouth along the thalweg at approximately 30 km." Modified

Line 133: Fix punctuation

Line 182: "Figure 3a shows various environmental parameters during the 2016 dry season." This sentence does not add much to the results. It is probably better suited as part of a figure caption.

We still think this sounds like a natural starting point before mentioning subtidal water level response to wind.

Line 182:  Figure 3a shows… For this sentence and similar ones. Construct the sentence so that you point out the results and then put the figure number at the end in parentheses example (Figure 3a) modified

Line 190:  remove "see" consider "(eg March ….) " modified

Line 251: remove "also" modified

Line 263: Define K1+O1. Also, define M2S2 in line 264.

This is clearly defined in Table 2, but we made some modification in the table as well. We think once the table is placed within the text it is much more readable.

Line 281: Replace "borrowing" with "using"

Line 282: Cut the second instance of "notation"

Lines 352-353: Consider the following sentence structure "Given recent advances in numerical methods and computational power, it has become possible to efficiently model the entire Mekong Delta in a combination 2DV and 3D." Done

355: Consider the following sentence structure "With this tool, 3D processes were fully represented, and salt transport mechanisms were modeled with 3-dimensional fidelity." Done

Line 359: forcings Done

Line 376: "… have a dominant role…" or "have dominant roles." Done

Line 391: No space before comma Done

Line 440: Remove "amongst others." Done

**Figures**

General suggestions: Many of the minor tick marks are not needed. Consider reducing the number of minor tick marks to improve clarity in several of the figures (for example X axes on Figures 3, 4,7, 8 and 9)

We respectfully disagree with the honourable reviewer. We believe, the minor ticks provide a tool for educational purposes, e.g., for younger scientists that aim to dive into details of the measurements or model results. At the same time, it is not a major distraction considering the size of the minor tick marks.

Figure 1 (b): Consider using a blue transect for Dinh An consistent with later figures. Done

Figure 2: Define HWS both panels show.  Added to the caption.

Figure 3: Point out Period 1 and Period 2 directly on the figure. Done

Figure 4: Bring the key up a bit so that it does not cover parts of the "a" plot. Sort the figures in order so that all the WL, then all the salinity plots, and finally all the discharge plots are together

We adjusted the legend but did not shuffle the panels, as the current separation is based on the two main river channels (Tien and Hau and their distributaries). The left panel is WL, Q and S for Tien River, and the right panel shows those of the Hau River.

**References**

Brakenhoff, L., Schrijvershof, R., van der Werf, J., Grasmeijer, B., Ruessink, G. and van der Vegt, M.: From Ripples to Large-Scale Sand Transport: The Effects of Bedform-Related Roughness on Hydrodynamics and Sediment Transport Patterns in Delft3D, J. Mar. Sci. Eng. , 8(11), doi:10.3390/jmse8110892, 2020.

Eslami, S., Hoekstra, P., Kernkamp, H., Trung, N. N., Duc, D. Do, Quang, T. T., Februarianto, M., Dam, A. Van and Vegt, M. van der: Flow Division Dynamics in the Mekong Delta: Application of a 1D-2D Coupled Model, Water, 11(4), doi:10.3390/w11040837, 2019a.

Eslami, S., Hoekstra, P., Nguyen Trung, N., Ahmed Kantoush, S., Van Binh, D., Duc Dung, D., Tran Quang, T. and van der Vegt, M.: Tidal amplification and salt intrusion in the Mekong Delta driven by anthropogenic sediment starvation, Sci. Rep., 9(1), 18746, doi:10.1038/s41598-019-55018-9, 2019b.

Gugliotta, M., Saito, Y., Nguyen, V. L., Ta, T. K. O., Nakashima, R., Tamura, T., Uehara, K., Katsuki, K. and Yamamoto, S.: Process regime, salinity, morphological, and sedimentary trends along the fluvial to marine transition zone of the mixed-energy Mekong River delta, Vietnam, Cont. Shelf Res., 147(March), 7–26, doi:10.1016/j.csr.2017.03.001, 2017.

McLachlan, R. L., Ogston, A. S. and Allison, M. A.: Implications of tidally-varying bed stress and intermittent estuarine stratification on fine-sediment dynamics through the Mekong's tidal river to estuarine reach, Cont. Shelf Res., 147, 27–37, doi:https://doi.org/10.1016/j.csr.2017.07.014, 2017.

Nguyen, A. D. and Savenije, H. H. G.: Salt intrusion in multi-channel estuaries: a case study in the Mekong Delta, Vietnam, HESS, 10, 743–754, 2006.

Savenije, Hubert H.G.: Salinity and tides in alluvial estuaries, 2nd ed., 2012.

Savenije, H. H. G.: Salt intrusion model for high-water slack, low-water slack, and mean tide on spread sheet, J. Hydrol., 107(1–4), 9–18, doi:10.1016/0022-1694(89)90046-2, 1989.

Tran Anh, D., Hoang, P. L., Bui, D. M. and Rutschmann, P.: Simulating Future Flows and Salinity Intrusion Using Combined One- and Two-Dimensional Hydrodynamic Modelling—The Case of Hau River, Vietnamese Mekong Delta, Water, 10(7), doi:10.3390/w10070897, 2018.

UNESCO: The Practical Salinity Scale 1978 and the International Equation of State of Seawater 1980, Unesco Tech. Pap. Mar. Sci., 36, 1981.

Vasilopoulos, G., Quan, Q., Parsons, D., Darby, S., Tri, V., Hung, N., Haigh, I., Voepel, H., Nicholas, A. and Aalto, R.: Anthropogenic sediment starvation forces tidal dominance in a mega-delta, Res. Sq., doi:https://doi.org/10.21203/rs.3.rs-81555/v1, 2020.